# Information acquisition and cognitive processes during strategic decision-making: Combining a policy-capturing study with eye-tracking data

Alice Pizzo[1]*, Toke R. Fosgaard[2], Beverly B. Tyler[3], Karin Beukel[2]

1 Copenhagen Business School, Department of Management, Society and Communication, Frederiksberg C, Denmark, 2 Department of Food and Resource Economics, University of Copenhagen, København, Denmark, 3 Department of Management, Virginia Tech Pamplin College of Business, Blacksburg, VA, United States of America

* api.msc@cbs.dk

**Data Availability Statement:** The data were now anonymized and made available on the platform Figshare at https://doi.org/10.6084/m9.figshare. 19753417.v1.

## Abstract

Policy-capturing (PC) methodologies have been employed to study decision-making, and to assess how decision-makers use available information when asked to evaluate hypothetical situations. An important assumption of the PC techniques is that respondents develop cognitive models to help them efficiently process the many information cues provided while reviewing a large number of decision scenarios. With this study, we seek to analyze the process of answering a PC study. We do this by investigating the information acquisition and the cognitive processes behind policy-capturing, building on cognitive and attention research and exploiting the tools of eye-tracking. Additionally, we investigate the role of experience in mediating the relationship between the information processed and judgments in order to determine how the cognitive models of student samples differ from those of professionals. We find evidence of increasing efficiency as a function of practice when respondents undergo the PC experiment. We also detect a selective process on information acquisition; such selection is consistent with the respondents' evaluation. While some differences are found in the information processing among the split sample of students and professionals, remarkable similarities are detected. Our study adds confidence to the assumption that respondents build cognitive models to handle the large amounts of information presented in PC experiments, and the defection of such models is not substantially affected by the applied sample.

## Introduction

Policy-capturing represents a prominent approach to study decision-making and strategic choices. An advantage of the method is that it allows researchers to infer what information has the most influence on respondents' assessments, judgments, and choices [1–3].

**Funding:** This work was supported by the Novo Nordisk Foundation [Grant #21630] - "Investigating the micro foundations of socioeconomic impact of university-industry relations", but the foundation was not involved in the study design, the data collection/analysis or the writing of the manuscript.

**Competing interests:** The authors have declared that no competing interests exist.

More specifically, policy-capturing is a methodology employed to assess how decision-makers use available information when asked to evaluate a hypothetical situation [4]. The purpose of policy-capturing (PC) is to capture individual decision-making policies, which reveal how they weigh, use and select information [5]. A policy-capturing experiment consists of repeated judgments in which respondents are asked to judge a series of simulated scenarios that are characterized by various degrees of attributes (information cues). The PC technique presumes that decision-makers in practice must make decisions based on more decision cues than humans can cognitively process [6]. Thus, their PC decision scenarios include more information cues in order to be more realistic [7]. They also suggest that respondents' evaluations in the judgment exercise can then be regressed on the variation in the attributes or information cues to determine what attributes of the decision scenario are significantly impacting the evaluation. The resulting coefficient estimates of the attributes indicate the relative importance of the attributes and provide an overview of the patterns and weightings used by the decision-makers, while avoiding the social desirability bias often associated with self-reporting [3].

The widespread use of policy-capturing (PC) experiments, in general, is highlighted in several methodological reviews [3, 8, 9]. PC, with a range of different design specifications, has been applied to investigate the decision rules in dozens of high-ranking publications in organizational and management research [3], and employed in a wide range of topics: job search [10, 11], compensation [12], employee discipline [13], job analysis [14], sexual harassment [15], employment interviews [16, 17], contract arbitration [18], motivation [4], promotion evaluations [19], financial investment judgments [20], and executive decision-making [2, 6, 21]. Moreover, PC has also been widely used outside organizational and management research in fields such as medical decision-making [22, 23], psychology [24, 25], and sociology [26, 27]. The present paper targets the assumptions and the underlying cognitive mechanisms of the specific design of policy-capturing developed by Hitt et al. [6] in 1979, applied in several papers of the organizational literature [2, 28–32].

An important assumption behind the policy-capturing technique is that as respondents review several scenarios one at a time, and make judgments, they develop cognitive models to help them process, interpret, and integrate the complex set of information provided in the policy-capturing experiment [6]. Therefore, each evaluation or judgement is the result of both the information supplied in the scenarios and the subjective cognitive models that participants bring with them and develop while participating in the exercise [2, 33]. This assumption builds on behavioral decision theory, which argues that due to bounded rationality [34–36], when processing large amounts of information, humans seek to reduce their cognitive effort through the formation of heuristics and preferences in order to exclude some available information in uncertain and complex contexts and simplify the decision process [20, 37–40]. Indeed, policy-capturing studies that include more information than humans can cognitively process have found that decision-makers approach ill-structured decisions with complex mental models to integrate the information into a single judgment [2, 6, 41, 42]. A similar assumption is foundational to the attention-based view of the firm, which recognizes that human beings have limited cognitive capabilities when processing all the available information that potentially is relevant for making decisions and judgements [43, 44].

Although PC has been prominent and its use widespread, the literature lacks an experimental investigation of the cognitive processes underlying the methodology. To circumvent that lack, this study builds on cognitive and attention research and exploits the potential of eye-tracking to study the process of answering a PC experiment. Additionally, we investigate the role of experience in mediating the relationship between the information process and the judgments to verify the reliability and appropriateness of the samples recruited. Indeed, Karren et al. [3] point out that despite being an important criterion in most research studies, few of

the reviewed PC papers have analyzed the reliability of their decision makers' judgments. This level of analysis relates to an ongoing academic debate on the use of students as experimental subjects [45–47]. Indeed, working with students as subjects, when leading to externally valid conclusions, can be a strategic decision in terms of availability, reachability, and costs. That is also the reason why we believe that not finding a substantial difference in the cognitive processes behind PC is a rather positive finding in the perspective of running policy-capturing studies on strategic decision-making with students as subjects. In sum, our research question is *what characterizes the cognitive processes of respondents undergoing a policy-capturing experiment*? *Are they affected by the applied sample composition*?

Policy-capturing and choice modelling methods share many similarities, as emphasized by Aiman-Smith et al. [9], and we therefore build our analysis upon the experimental approaches previously applied to choice modelling studies, such as in Meissner et al. [48] and in Hoeffler et al. [49]. It is important to note that differences do exist between the methods, as such the number of attributes the respondents are exposed to, whether the extracted rating is continuous or not, and whether the attribute selection is theoretically driven or empirically driven. Aiman-Smith et al. [9] identify several analogies in various disciplines that use similar regression-based methods for investigating decision-making. For example, traditional conjoint analysis is the preferred method in marketing research, contingent preference is prominent in environmental and social policy research, and policy-capturing is more common in strategy and human resource management studies. Given the strong similarities, the methodologies employed in choice modelling can be considered useful guidelines for researchers applying policy-capturing [3, 50]. Moreover, also research based on choice modelling has used eye-tracking techniques to explore how participants process information in order to test the assumptions of choice experiments [51–61]. Prior work at the intersection of attention and choice modelling contributes to our four *hypotheses* about how participants complete the PC experiment. The hypotheses, further discussed below, are that respondents become more selective, more consistent, and more efficient over the course of the experiment, and, finally, that experience also affects these cognitive processes.

We have completed a PC experiment and measured the participants' attention processes with an eye-tracking device. The topic of the PC experiment is inter-organizational collaboration, but in the present paper, we exclusively report on the processes of answering the experiment, not on the measured drivers of inter-organizational collaboration. We have designated an entire sister paper to address the evaluation outcomes of the PC experiment [62]. We will merely connect the answers in the PC experiment to attention. Doing this, we find evidence that respondents of a policy-capturing study build cognitive models to cope with the large amount of information provided. More precisely, the effort, measured as a function of time spent looking at the information that needs to be evaluated, decreases with practice, signaling an increase in efficiency. Furthermore, a selection of attention also emerges among the different attributes that characterize an evaluation scenario. The selection is associated with the respondents' evaluation of the PC scenarios. The applied sample, and hence experience, is found to have little influence on the detected cognitive models, although a few differences do apply. The student sample is quicker in reducing the effort, but they are not systematically diverging in their attention patterns. They spend a similar amount of time on scenarios, and they show similar patterns of attention, which undergo a selection process for both. Interestingly, both samples are found to show consistency in what they weigh to be important and what they state it is.

The paper is structured as follows. In the next section we build the theoretical framework upon which the empirical analysis is based, in the following section we develop the hypotheses. We move on to a discussion of the methods, present the results, and end with a discussion of our results, limitations, and conclusions.

## Theoretical framework

The attention-based theory developed by Ocasio [43] defines *attention* as the "noticing, encoding, interpreting, and focusing of time and effort" on the relevant information processing to evaluate choices and make decisions. Attention itself represents a good indicator of the limited information processing capability of humans [63]. In situations with more information to be processed than what is cognitively possible for decision makers to handle, *selective attention* is described by Ocasio [44] as a "process by which individuals focus information processing on a specific set of sensory stimuli at a moment in time". Subjects choose which stimuli to attend to and which to screen out [64].

The close link between attention and decisions is dissociated from the traditional economics assumption, which states that every decision maker opts for a certain choice based on pre-existing preferences and all the available information [65]. In behavioral decision research, the *constructive preferences* approach argues that preferences are constructed by the decision maker within the specific task and context of a decision [66, 67]. The availability of information in a given context and the features that characterize the contextual environment are essential determinants of the constructive preferences.

Attention-based theory assumes a similar process of decision-making as the one described above [43]. While the decision context relates to the information that needs to be evaluated and processed by the decision maker, some subjective characteristics are also influential factors in the cognitive process. Age, gender, type of education, and nationality are some of them. Experience is another subjective characteristic. In the organizational domain, in accordance with the stream of research on long-term working memory [68], respondents with experience are found to encode and retrieve information more rapidly than inexperienced respondents and are able to efficiently access the knowledge acquired later. Therefore, the cognitive process that respondents undergo during a policy-capturing experiment, composed of simulated scenarios related to the working place and strategic organizational decisions, might be moderated by their level of professional experience [69]. Scholars have also found that experience is related to the *information-reduction hypothesis* [70]: expertise allows respondents to direct more selective attention to stimuli that are relevant to their decision-making [71]. Thereby, attention is allocated more efficiently by experienced respondents.

Finally, relevant to the present study is the effort to combine eye-tracking with choice-based research to explore to what extent respondents make coherent decisions: they might be influenced by the survey context, information cues, ordering effects, and their own experience as reflected in their demographic characteristics. Several studies have combined choice modelling experiments with visual attention data to test the choice experiments' undergoing cognitive processes [40, 51–56, 72, 73]. Scholars have indeed applied eye-tracking as a measure of attention to quantify individuals' information processing in choice-based exercises in different domains such as decision-making in economics [51, 56, 74, 75], consumer choice [48, 52, 75–81], medical decision-making [54, 82], and food choice in the sustainability field [83–85]. In a comprehensive review of eye-tracking, Orquin [86] explains how choices and attention are interrelated, while Ashby et al. [87] provide a review of the reasons why the use of eye-tracking methodologies has increased in the field of behavioral decision-making. The present study seeks to combine the measurements of eye movements and the lessons learned for choice-based models to investigate the cognitive processes underlying this policy-capturing technique.

## Hypotheses

In our study we seek answers to four hypotheses related to the cognitive processes underlying the policy-capturing technique, by combining the PC experiment with eye-tracking.

Eye-tracking has been identified as a tool that can inform about the process of answering top-down assigned tasks, such as PC studies [86–88]. The idea is that tasks and goals of an experiment should motivate respondents to use deliberate reasoning to analyze opportunities, within their information processing cognitive constraints, to make up their judgements [37, 89–91]. Visual attention represents the psychological construct of focus in eye-tracking research. The notion behind the practice of quantifying attention is the so-called *eye-mind assumption*, which assumes a tight link between what is seen and what is cognitively processed [92]. Successively, the *relative eye-mind hypothesis* was developed by Huettig et al. [93] who specified how the most active part in the working memory will eventually determine the likely direction of the eye movement for any given moment in time.

The use of eye-tracking has provided a better understanding of how visual design of a task shapes attention [94], how attention develops with the repetition of a certain task [86], and how accumulated attention affects the cognitive models that predict decisions [95]. More specifically, scholars have identified that the number of *fixations* and the *time spent*, two of the basic eye-tracking metrics, on a specific area of interest is an indicator of how much attention the individual directs to that area [96, 97].

## Efficiency

To learn about the cognitive processes underlying a PC experiment, we investigate the development of *efficiency*, which can be measured as the reduction in attention found with repetitions of the same task [77, 98–100]. Especially in repeated tasks, respondents get better at extracting information and retaining it over time [87]. If the respondents are asked to repeat the same decision task multiple times, it is expected that the amount of effort required to perform the evaluation will reduce over time since respondents take their decisions based on previous choices, internal decision rules, or rules of thumbs [49, 101]. Because eye-tracking research has found experimental respondents to spend progressively less time looking at the information provided before making a judgement in later repetitions than they do in early repetitions [102], a similar decreasing path of attention is expected to emerge. Hence, we hypothesize the following.

*H1.1*: *respondents of a PC exercise become* **more efficient** *in directing attention to attributes with practice.*

We test this hypothesis by using the eye-tracking measure of *time spent* as in Hoeffler et al. [49]. Time spent is the total amount of time devoted to the available information in each scenario.

## Selectivity

Because in a policy-capturing exercise the availability of the material provided in each scenario exceeds the amount of information that respondents can process [29], we expect that respondents will assess the policy-capturing exercise based on only a *selection* of attributes and that those selected attributes are more important for their answer in the PC experiment. This conjecture is consistent with the attention-based literature, the findings from eye-tracking research, and its applications on choice-based studies. Thus, the hypothesis is summarized as follows.

*H1.2*: *respondents become* **more selective** *in the information acquisition with practice during a PC exercise.*

To test this hypothesis, we focus on the eye-tracking measure of *fixations* as in Meissner et al. [48]. A fixation is defined as the dwelling of attention on a certain piece of information.

## Consistency

*Consistency* is a key feature of the information acquisition process behind any repeated choice exercise, as it opens for predictability [48, 49]. We are interested in investigating the relationship between what respondents do while answering a PC exercise (in other words, what mental schemes they apply or how they weigh attributes) and what they state they do (i.e., what they think and say is important for their decisions).

*H1.3*: *respondents are **consistent** between what they weigh as important and what they say is important.*

Despite including eye-tracking data in the analysis, we investigate this matter mainly by comparing respondents' evaluation extracted from the completion of the PC experiment and a self-reported evaluation where the respondents rank the available information from the most to the least important [49].

## Experience

We are also able to investigate the filtering effect of experience thanks to the sample composition of the experiment. Indeed, we recruited eighteen professionals working in the science industry and twenty-six MSc students with training in science. The impact of *experience*, intended as the number of years of professional experience after the most recent degree, is analyzed across the split sample of students and professionals, by the means of statistical estimations from the PC outcomes and the eye-tracking metrics. Because students on average have less professional experience than professionals working in the industry (1.84 years vs. 11.19 years of working experience), their ability to make judge about inter-organizational collaboration is likely limited [103]. Both the organizational and the experimental literature have suggested that there exists a difference between actual workers in the field and students in the lab, since experience plays a role in experiments [104]. On this premise, the second level of analysis of this research converges to the following hypothesis.

*H2*: *experience affects the cognitive processes behind a PC exercise.*

To test this, we replicate all previous evidence by focusing on the sample composition. On one hand, experience is expected to have a boosting effect on efficiency because professionals are more familiar with the evaluation context of the PC exercise, therefore the required effort to fill in the survey can be expected to be lower. On the other hand, the effort dedicated to the task could be higher for experienced respondents, since students have less sense of the collaborations that they are asked to evaluate and might skip over the exercise faster.

## Materials & methods

### The policy-capturing decision models and procedure

In this study, policy-capturing tool was used to investigate industrial scientists' assessment of potential collaborative opportunities with academics [6, 30, 31]. We created a policy-capturing survey using the online survey software *Qualtrics* to determine the attributes that scientists focus on and weight more heavily when they evaluate research collaboration opportunities with university academics. The *Qualtrics* policy-capturing exercise included two instruments which alternate randomly across respondents: 1) the policy-capturing block, which consisted

of an instruction page, 30 randomly ordered pages, each describing a collaboration decision scenario, and a final page where participants rank ordered the importance of the decision attributes used in the scenarios; 2) a background survey collecting information on participants' demographics and attitudes. While the first block includes individual direct questions to be answered on a Likert scale, the second block constitutes the actual experimental task—the policy-capturing exercise.

**1. The policy-capturing instrument.** The first page of the policy-capturing instrument provided participants with instructions for the exercise and showed them the two questions they would be asked after they reviewed the information included in each scenario. The 30 scenarios that followed the instructions described potential collaborations consisted of 16 decision attributes that were randomly given different degrees of weight, and two questions asking participants to rate the attractiveness of the collaborative opportunity. We formulated 16 attributes to describe each of the 30 scenarios of potential collaborations with university academics. The number and the design of the scenarios were set-up by following the methodology as in Hitt et al.; Tyler et al. [6, 29]. More specifically, Hitt et al. [6] were the authors who in 1979 developed the specific design of the policy-capturing (PC) questionnaire applied in several papers thereafter. The specificity of this version of PC concerns the number of the repetition of the scenarios, the number of the attributes included and repeated in each scenario, the randomization of the order in which the scenarios are administered to the respondents versus the iteration of the same orders of the attributes (randomized only once during the design phase). The application of the policy-capturing methodology, as develop by Hitt et al. [6], while providing a close reference to prior and well-established literature, made us preserve the key feature of the questionnaire design. The attributes of the PC exercise have been formulated in accordance with the constructs of four relevant organizational theories extracted from the organizational literature. Specifically, the four theories were identified within those considered important for collaborations between professional and academic scientists: transaction cost economics [105], resources-based view [106], regulatory focus theory [107, 108] information economics [109]. An overview of the 16 attributes can be found in Table 1 and the exact layout used and repeated verbatim over all 30 scenarios can be seen in S1 Fig, which shows a screenshot of one selected scenario.

Prior to conducting the study, we organized a focus group with a number of junior and senior researchers in the organizational area at the local university to test the representativeness of the attributes. Moreover, feedback from senior experts in the use of policy-capturing exercises were sought. Although the policy-capturing questionnaire was designed to assess the influence of the sixteen attributes suggested by the four theories, the weighting by the respondents of these attributes is not the main focus of analysis, but it rather serves as a tool to investigate how the process of decision-making unfolds.

The attributes were displayed in the same graphical order, while the degrees characterizing the 16 attributes, distributed on a five-point Likert scale—ranging between low, moderately low, average, moderately high, high, were randomly assigned for each of the scenarios. We therefore end up with 30 unique scenarios, identified as different opportunities with different degrees of the describing attributes. The mixture of elements in the 30 scenarios were determined from a fractional factorial design. To further verify the independence of the attributes, the correlation of the attribute degrees within and across theory was tested and found to be ranging between ±0.45. For each scenario, the respondents were asked to evaluate the attractiveness of the collaborative opportunity by responding to two questions on a seven-point Likert scale ranging from 1 ("very unattractive/very low") to 7 ("very attractive/very high") (31). The two evaluation rankings were kept the same over the thirty scenarios and were formulated as follows:

**Table 1. Theoretical constructs.** Overview of the theoretical constructs behind every attribute of the PC scenarios.

| Theory | Construct | Attribute Text |
|---|---|---|
| Transaction Costs Economics | Asset Specificity | *Level of investments in equipment required for this project that cannot be used in other research projects (i.e., investments that cannot be transferred to other collaborations).* |
| | Small Numbers | *Number of other partners currently interested in cooperating with you.* |
| | Formal Governance | *Extent to which this collaboration will be coordinated by and controlled by detailed contracts.* |
| | Informal Governance | *Favourability of the collaborative partner's cooperative history.* |
| Information Economics | Asymmetric Information | *Disciplinary overlap between your technical knowledge and that of the other partner in this collaboration.* |
| | Asymmetric Information | *Degree to which the other partner in this collaboration possesses intangible assets that are difficult for you to value.* |
| | Adverse Selection | *Your familiarity and knowledge of the collaborative partner's knowledge, skills and capabilities.* |
| | Adverse Selection | *Extent to which the collaborative partner's co-authors and colleagues are considered to be reputable.* |
| Resources Based View | Financial Resources | *Financial resources the collaborative partner's organisation is committing to support the collaboration.* |
| | Human Capital Resources | *Extent to which the collaboration provides you with access to valuable, rare intellectual talents.* |
| | Physical Capital Resources | *Extent to which the collaboration will give you access to valuable, rare equipment.* |
| | Imitability | *Degree to which the intellectual capital created in this research collaboration will be openly and broadly shared.* |
| Regulatory Focus Theory | Prevention & Emotional | *Extent to which not meeting goals on this collaboration will undermine future collaborations.* |
| | Prevention & Vigilance | *Need for partners to diligently/constantly search for problems and difficulties during the collaboration.* |
| | Promotion & Behaviour | *Extent to which future activities in the collaboration will be decided by the partners based on intermediate outcomes.* |
| | Promotion & Relational | *Degree to which this collaboration can be expected to establish a close, trust-based relationship between the researchers.* |

*"Based on the information provided above and your experience, please rate the attractiveness of this collaboration?"*

*"Based on the information provided above and your experience, what is the probability that you would further explore this collaboration?"*

The *Qualtrics* software also randomly determined and recorded the order at which the 30 scenarios appeared to the respondents. Hence, using the degrees of the attributes that characterized each simulated scenario as the independent variables and the resulting evaluation rating completed by the respondents as the dependent variable, regression analysis could be performed to reveal the respondents' decision models [6]. S1 Fig provides a graphic overview of what is meant by degrees of the attributes that characterized each simulated scenario. In short, respondents were instructed to provide an evaluation of 30 scenarios, which are described by a list of 16 attributes (located in the same position in each scenario) that vary within a certain range of values (five degrees from "low" to "high") in a random fashion. While the position of the attribute is the same for every scenario, its characterization (its degree) changes randomly across scenarios. All respondents eventually answer to the same 30 scenarios but each in a different random order.

The final page of the policy-capturing instrument asked respondents to rank order the sixteen attributes used to create the 30 scenarios according to the respondent's perceived importance.

**2. Additional survey.** In addition to the policy-capturing instrument, each respondent completed a survey consisting of demographic data, and a part with a number of scales measuring attitudes and preferences related to their work environment and professional life. The material included in the online survey consisted of these two blocks, which were randomly ordered along with the policy instrument by *Qualtrics* for each respondent to control for potential order effects.

## Eye-tracking measures

The eye-tracking equipment enabled us to collect evidence to study the cognitive process that respondents of the policy-capturing go through. Eye-tracking is a common technology to measure where someone is looking or how they visually scan a specific situation. Eye-tracking measures are a useful tool for both qualitative and quantitative research, as it allows researchers to tap into non-conscious processes including biases, heuristics, and preference formation [110].

Visual attention, also defined as "selectivity in perception", is performed in people by moving their eyes around the situation at display (screen, text, graphic material, or choice scene). Since only 2% of the visual area is projected into the *fovea*, the central part of the retina at high density of sensory neurons, the eyes need to move in order to inspect stimuli and fully acquire information [111]. Technically speaking, the eye-tracking equipment calculates the location of the respondent's *fixations* and *gaze points*, which are the basic output measures of interest and the most used terms. Gaze points show what the eyes are looking at. Our eye-tracker, with a sampling rate of 60 Hz, can collect 60 individual gaze points per second. If a series of gaze points is very close in time and space, the gaze cluster constitutes a fixation, denoting a period where the eyes are locked towards an object. As mentioned above, a *fixation* is defined as the instance in which the eyes are stably resting on a certain stimulus. Instead, the quick movement of the eyes between different consecutive fixations is called a *saccade* [112]. Researchers, such as Rayner [111], have demonstrated that information acquisition happens only in correspondence with a fixation, and not during saccades. For this reason, a number of eye-tracking measures are based on fixations as the unit of analysis. The duration threshold in milliseconds (ms) to identify a fixation is 100 ms. Such threshold was not deliberately decided by the authors but employed as part of the data segmentation of the eye-tracking software used for the study (iMotions A/S, version 7.1). In iMotions, if the duration of a fixation is less than 100 ms, it gets discarded. Only fixations candidates that are longer than 100 ms count as a fixation. Moreover, the eye-tracker collects the high frequency attention data of the entire period in which respondents are participating in the study. To organize the massive amount of data collected, some *Areas of Interest* (AOI) were defined to better summarize the attention data. More specifically, the areas were divided into three main groups: The attributes—The degrees—The answers. Attention on every single group can be analyzed separately. This structure enables to discern the composition of each scenario and to gather eye-tracking evidence at the needed level of details. S1 Fig shows the structure of the AOIs for one randomly selected scenario from the policy-capturing experiment. The eye-tracking measures, such as gazes, fixations, revisits, etc., recorded in the AOIs were used in our analysis. A brief explanation of these common eye-tracking measures follows below.

The metric *Time Spent* (briefly introduced above) quantifies the amount of time in milliseconds (ms) that respondents spend looking at a particular item or AOI. As for prior literature,

spending more total time looking at a specific piece of information is an indicator of preference for making a decision that is consistent with that information [58, 113, 114].

The variable *Fixation* count the number of fixations registered in a specific AOI; its interpretation is similar to time spent but the unit of measure (counts, not time) and its order of magnitude serves as a test case to quantify attention.

### Eye-tracking set-up

To collect eye-tracking data, we used a remote *Tobii X2-60* eye-tracker magnetically attached to the bottom of a monitor, in an on-sight room reserved for the experiment. In addition to the monitor, a mouse and a keyboard were made available to the respondents to indicate their decisions in the study. The 24-inch monitor had a resolution of 1920 x 1080 pixels, and it was connected to a computer behind a partition, where a researcher controlled the software (*iMotions A/S*, version 7.1). The survey was built on the *Qualtrics* platform and connected to iMotions as a plug-in. Before starting the experiment, respondents received information about the procedure of the study, the content of the survey and further instructions regarding the use of the eye-tracker. Once the respondents had asked their questions, the eye-tracker was calibrated on a 9-point calibration to ensure sufficient precision of the tool: the calibration outcome was considered to be positive if the mean difference between the measured gaze data and the target point was under or equal 40 pixels. Respondents completed the study one at a time, with a researcher present in the room operating the eye-tracking and no time limit.

In relation to data collection, the software iMotions combined data from the survey and from the eye-tracker in synchro. iMotions automatically segmented eye-tracking variables (e.g., number of fixations, amount of time spent, number of gazes, time to first fixation, hit time, number of revisit fixation duration). The eye-tracking data were stored on iMotions in the lab computer, while the policy-capturing decision data were stored online on Qualtrics and eventually downloaded. The two datasets were separately analyzed, then made compatible and merged for common analysis. Thanks to the online link plug-in function of iMotions, it was possible to overlap and temporally align what was showcased on the computer screen and the data collected by the eye-tracker.

### Pilot

A pilot study was conducted to verify the initial structure of the policy-capturing survey and to pre-test the technical set-up of the eye-tracking equipment (including both the respondents' monitor and the researcher's workstation of the eye-tracking setup). A researcher recruited twenty-eight students, by presenting the project to two Master of Science classes and by encouraging students to participate in a pilot of the experiment in October 2018. The students made individual appointments to participate in the pilot study in a departmental lab on campus. In the lab, the researcher provided each respondent with a simple overview of the study, and requested they sign a written consent form. No minor was included in the study. The form was safely stored by the researcher. Participants signed the consent in front of the researcher. Moreover, respondents had to agree to the terms and conditions on the first page of the *Qualtrics* program in order to proceed to the survey. The results of the pilot study were used to improve the experimental materials.

### Sample

To implement the study, we obtained a sample composed by students and professionals with training in science. To solicit science professionals, between January and April 2019 we contacted four companies in sectors representing science in the local area. Each manager

identified and invited key employees to participate in the study, and then provided us with a list of names of those willing to participate. We received the names of forty-six professionals who were e-mailed a request to participate in the study and were given two options: answering to the policy-capturing study on their personal computer or on a computer in our mobile on-site eye-tracking lab. While twenty-two respondents selected the first online option, eighteen respondents showed up in the on-site lab for individual appointments and participated in the policy-capturing while being eye-tracked. Students in a Master of Science Entrepreneurship and Innovation class were also asked to participate in the study in November-December 2018. To reduce potential attrition and to balance the incentive that professionals might have because of their managers' influence, students were told that one of them would be randomly drawn to receive a cash prize of approximately 60 euros. Thirty-two students agreed to participate, however only twenty-six completed the experiment. Ten of the students were enrolled in a Food Innovation and Health program and sixteen were enrolled in a Food Science and Technology program. Both subsamples took the policy-capturing exercise in the eye-tracking setting. When the respondents went to the department eye-tracking lab, the same procedure as for the professionals was followed: they participated one at the time, were given a brief overview initially, signed the informed consent for the eye-tracking experiment, and clicked the required box on the terms and conditions for the policy-capturing study. No minor was included in the study. Participants signed the consent in front of the researcher, who safely stored the form. Moreover, respondents had to agree to the terms and conditions on the first page of the *Qualtrics* program in order to proceed to the survey. The twenty-six students provided complete data and one of the students was randomly selected to receive the cash prize. Thus, our results report data collected for twenty-six students and eighteen professionals. The sample size is in line with policy-capturing studies of reference [6, 29]. The average age of the combined sample was 36 years old (25.5 for the student sample and 42.9 for the professionals), and the percentage of males was 36% (19% for professionals and 61.5% for students).

The policy-capturing exercise was performed during the academic year 2018/2019 as part of a funded project at the local university. Several academic outcomes, with distinct scope, level of analysis and research focus, have been developed by exploiting the data collected during that time; as a result, the methodological description is similar and shared with the method section of our other current working paper. In accordance with Danish legislation, there was no need for an institutional review board approval (IRB) for this study, since sensitive data—as defined by the Danish Data Protection Agency—was not retrieved from participants. The study did collect written consent from the participants in line with good ethical research practice. Written and digital consent was obtained. The complete dataset, without any identification of the participants, is posted on the *Figshare* platform [115].

## Results

Our empirical strategy is based on previous work that investigate the cognitive processes behind decision-making. Specifically, we apply the constructs and the analytical framework from Hoeffler et al. [49], who study the process of constructing stable preferences, which is relevant for our hypotheses on efficiency and consistency. Furthermore, we also use the framework of Meissner et al. [48], who study the role of attention in choice tasks, which is relevant for our hypothesis on selectivity. We decided to opt for this approach to seek legitimization in applying eye-tracking to investigate the policy-capturing methodology, which, to the authors' knowledge, was never done before.

For clarity, we begin by defining the terminology we use in the rest of the paper. We provide a short definition of each construct used in the paper. We define an *attribute* as one of the features

that characterize a certain alternative [48], which in our case is the attributes that characterize the 30 policy-capturing scenarios that respondents are asked to evaluate. We describe *effort* as the amount of time that respondents invest in delivering an answer, measured as their response time [49]. *Efficiency* is the process in which decision-making progressively requires less effort to reach a resolution of a task: it is closely related to effort, that is also defined as the amount of mental energy required to make up one's mind [48]. *Selectivity* refers to the process of selecting to attend on progressively smaller amounts of information. We refer to *preferences* as the relative importance respondents place on attribute both during the PC evaluation and the final ranking exercise. *Consistency* in decision-making relates to a non-contradictory pattern of choices. It is different from what Hoeffler et al. [49] call *violations*, which can be defined as the magnitude of the mismatch between two rankings (in our case, between the PC attributes ranking and the rated attributes ranking). *Experience* is professional expertise, understood in this policy-capturing study as the number of years of professional experience after the most recent degree of education. Despite efficiency and selectivity both representing a measure of attention, we believe that they play two important roles in completing our analysis respectively. On one hand, the two concepts are applied independently from each other and test different mechanisms, which we believe justifies reporting both. On the other hand, two different eye-tracking measures are employed to study efficiency and selectivity, as explained in detail in the section below.

Our results on the respondents' decision processing are based on three different sources of information:

1. *PC data*: the evidence originated from the policy-capturing evaluation exercise. We use answers about collaboration in the 30 policy-capturing scenarios to assess what pieces of information respondents weigh in their evaluation.

2. *Ranked data*: the evidence provided in the rating task at the end of the PC assessment. The individual respondent ranks the elements of the collaboration scenarios from most to the least important for their assessment. It was not possible to repeat a ranking position.

3. *Eye-tracked data*: the evidence originating from the eye-tracking technology. We select the attention data that is relevant for our analysis, the AOIs. In particular, we select attention on the *attributes* and on the *degrees* in the collaboration scenarios.

## Efficiency (H1.1.)

Our starting point for investigating the information acquisition process behind the policy-capturing method is to check whether respondents become more efficient throughout the evaluation process. The idea is that, facing the massive amount of information in the PC, the decision maker aims to be as efficient as possible in evaluating the choice scenarios by using the most important information while limiting the time spent overall. During what Hoeffler et al. [49] call the *constructive preferences* building phase, cognitive effort decreases, and efficiency increases accordingly. Because every choice is a process following which preferences are consolidated to then land at a resolute assessment [49], we expect to find a decreasing pattern of attention allocated to the attributes and degrees over time. A proxy for effort, and therefore for efficiency, is the time spent, calculated as the average times spent on respectively the attributes and the degrees. On average, respondents of our PC exercise dedicate 15.58 seconds to look at the attributes in each scenario, and 4.76 seconds to the degrees in each scenario. The development of attention over time is shown in Fig 1, which also includes the split between Professionals and Students.

Fig 1 shows the logarithmic form of the average time, in seconds, per scenario spent on respectively all the attributes and all the degrees. A clearly descending pattern can be seen for

the attributes, while attention on the degrees is at a stable and lower level. The difference between the two types of AOIs is naturally affected by the content and the size of the area: the attributes describe in words the characteristics of the scenario; the degrees describe, with a simple cross on a scale, the extent to which the attribute applies. See S1 Fig for reference. We find a significant reduction in average time spend on the attributes in the first ten scenarios compared to the last ten (the difference is 12.16 seconds, a pair t-test p<0.001). For the degrees, the same comparison is also significant, but smaller of magnitude (the different is 12.46 seconds, p = 0.015).

We also test the efficiency on the attributes and the degrees econometrically using a random effects regression clustered at the individual level (Table 2). We use a random-effect model due to the panel-data nature of the attention data over scenarios.

We treat the average time spent as dependent variable and use a log-transformation to better model the nature of the data. The variable *round* (ranging from 1 to 30 according to the individual chronological order of the scenarios) is treated as an explanatory variable. We add several demographic controls (i.e., *gender, nationality, educational level, working experience*). We find that *round* is highly significant, regardless of the control variables being added or not (t = -0.064; p<0.001), suggesting a systematic reduction in effort (i.e., an increase in efficiency) over the repetition of the scenarios. To have a more relative measure of attention and as a sensitivity check, we replicated the same analysis as above (Fig 1 and Table 2) with an individual measure of time spent on attributes and degrees as the proportion of the total time spent on all 30 scenarios for each participant: the same results hold.

The rich data structure allows to link the process of the decreasing effort to both the importance respondents seem to put on the attributes according to the PC data and the importance they reveal in the Ranked data. For the PC data, we regress the respondents' answers of the collaboration opportunity across all scenarios on the degree (the value) of each of the attributes. Based on these individual regressions, we extract whether attributes are significant or not. We then compare the reduction in effort (decrease in average time spent on the first ten compared to the last ten scenarios) with the significance dummy from the PC data in a t-test. We do not

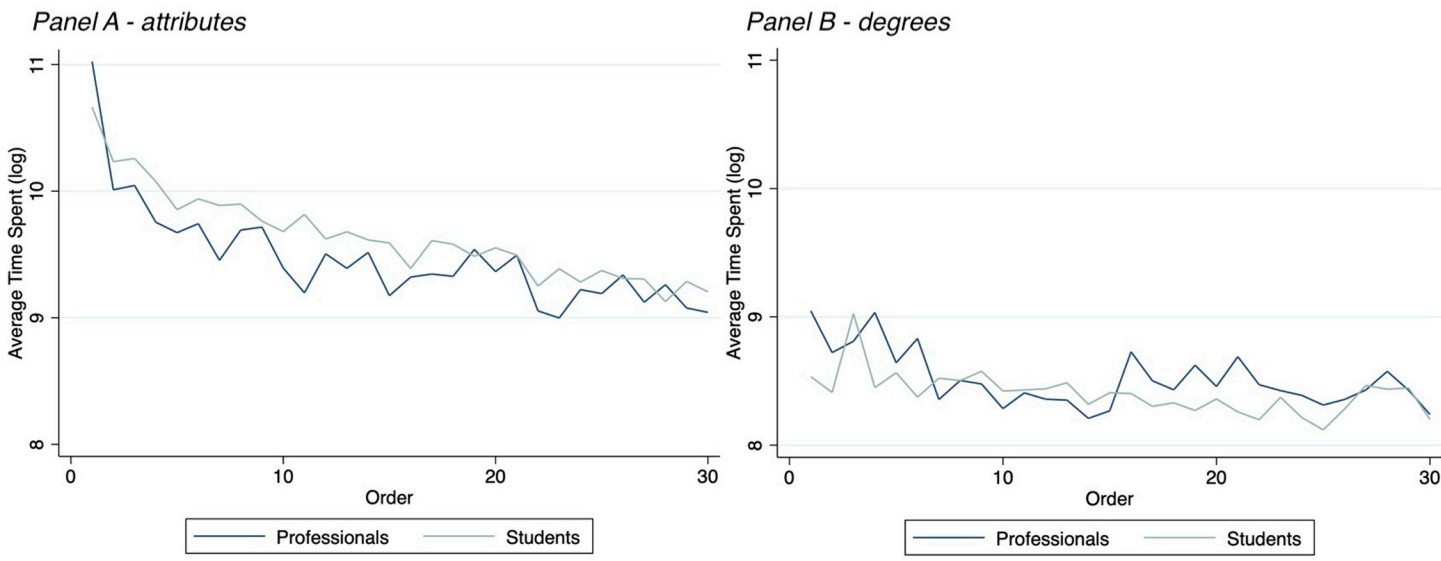

**Fig 1. Attention.** Pattern of attention measured in time spent (seconds) for attributes (Panel A) and degrees (Panel B) over the 30 repeated collaboration scenarios, by professionals and students. The logarithmic form of the variable of interest is depicted in the graph.

**Table 2. Regression table H1.** Random effects regression models for both attributes and degrees covering hypothesis 1.1 and hypothesis 1.2 (p-values within parentheses).

| | Hypothesis 1.2—efficiency | | | | Hypothesis 1.2—selectivity | | | |
|---|---|---|---|---|---|---|---|---|
| | for attributes | | for degrees | | for attributes | | for degrees | |
| Rounds | -0.064*** | -0.064*** | -0.022*** | -0.022*** | 0.122*** | 0.122*** | 0.042* | 0.042* |
| | (0.000) | (0.000) | (0.000) | (0.000) | (0.000) | (0.000) | (0.030) | (0.030) |
| Gender | | -1.022 | | -1.339* | | 1.704 | | 1.685 |
| | | (0.145) | | (0.048) | | (0.192) | | (0.184) |
| National | | 0.497 | | 0.627 | | -1.184 | | -1.326 |
| | | (0.446) | | (0.320) | | (0.337) | | (0.268) |
| Education | | 0.257 | | 0.240 | | -0.517 | | -0.568 |
| | | (0.380) | | (0.397) | | (0.338) | | (0.282) |
| Years of work experience | | -0.015 | | -0.031 | | 0.068 | | 0.104* |
| | | (0.730) | | (0.481) | | (0.156) | | (0.041) |
| Constant | 9.499*** | 8.584*** | 7.800*** | 7.116*** | 3.804*** | 5.685* | 6.443*** | 8.437*** |
| | (0.000) | (0.000) | (0.000) | (0.000) | (0.000) | (0.014) | (0.000) | (0.000) |
| R-squared | 0.044 | 0.082 | 0.006 | 0.080 | 0.047 | 0.094 | 0.006 | 0.078 |
| N | 1320 | 1320 | 1320 | 1320 | 1320 | 1320 | 1320 | 1320 |

* $p < 0.05$,

** $p < 0.01$,

*** $p < 0.001$

d.v. log(time spent per scenario)

d.v. sum of no fixation attributes per scenario

find a significantly different reductions in efficiency across the attributes found in the PC data as significant as opposed to non-significant for either attributes (t = 0.262; p = 0.396) or degrees (t = 0.514; p = 0.303).

For the Ranked data, we take a similar approach. We label the highest ranked attributes as important. To make the PC data and the Ranked data comparisons parallel, we allow the same number of attributes to be marked as important, based on the number of significant attributes that each individual shows in the regression. Again, we find no significantly different reductions in efficiency across the attributes ranked highest as opposed to those ranked lowest, either for the attributes (t = -0.190; p = 0.424) or the degrees (t = 0.338; p = 0.367). The results indicates that the decrease in effort, the efficiency gain, is not different across the attributes identified as more important in the PC data and Ranked data.

While our findings suggest that the respondents' effort is decreasing, and that the respondents become more efficient, the process seems not to be associated with the attributes identified as most important for the individual respondent.

## Selectivity (H1.2)

We also explore how our respondents decide to distribute their attention. Generally, selection occurs when attention is pulled to the attributes that represent the most important piece of information to the individuals to make their assessment [48]. The layout of the thirty policy-capturing scenarios facilitates the respondents to detect and focus on the most crucial attributes, and the associated degrees. Indeed, the position of the attributes in the scenario does not change over the repetition of the task. We find that participants on average attend to the first half of the attributes graphically displayed on the screen faster than the second half

displayed on the lower part of the screen (time to first fixation is on average respectively 23.66 and 31.10; p<0.001), suggest that a top-down process applies.

Moreover, because of the structure of the scenarios, it is an easy task for the researchers to identify the location of the most important attributes and degrees. We operationalize the selection by coding for each scenario, which of the attributes and the degrees respondents did not fixate on.

Fig 2 displays the average number of attributes receiving no fixations per scenario over time. Each bar of Panel A is calculated as the sum of *attributes* receiving no fixations within a scenario, averaged among all respondents, while Panel B the same measure for all *degrees*. Panel A of Fig 2 describes an increasing pattern of zero-fixations on the attributes starting at 2 in the first scenario, growing to 6, to then stabilizing at around 7 in the last scenarios; panel B shows a more constant trend that starts at around 6 zero-fixations and only reaches 8 by the end of the PC exercise. A paired t-test comparing the average zero-fixations of the first ten scenarios with those of the last ten finds a significant difference for attributes (p<0.001), with additional 2.37 attributes receiving no attention on average. For the degrees, the difference is also significant (p = 0.037), with 0.72 additional degrees not being fixed at, on average.

We add to this evidence two random effects clustered regressions with the variable *round* as explanatory variable and the sum of attributes with no fixation for each scenario as the dependent variable. Additionally, we control for demographics variables: *gender*, *nationality*, *educational level*, *working experience*. The *round* variable is significant for both the attributes (t = 0.122; p < 0.001) and the degrees (t = 0.042; p = 0.044), suggesting that an attention selection process occur during the policy-capturing exercise. As a contrast to the zero-fixations, we also analysis what is in fact fixated on. A paired t-test comparing the average fixations of the first ten scenarios with average fixations in the last ten scenarios finds a significant difference for attributes (p<0.0001), with 28.8 fewer fixations on average. For the degrees, the difference is also significant (p = 0.0018), with 3.1 fewer average fixations.

Another level of analysis concerns whether attributes and degrees are viewed in isolation, meaning that one is viewed while the corresponding counterpart is not. In other words, we calculate the percent of attributes that are viewed without examining their degree, and the number of degrees that are viewed without examining their attribute. On average, the attributes

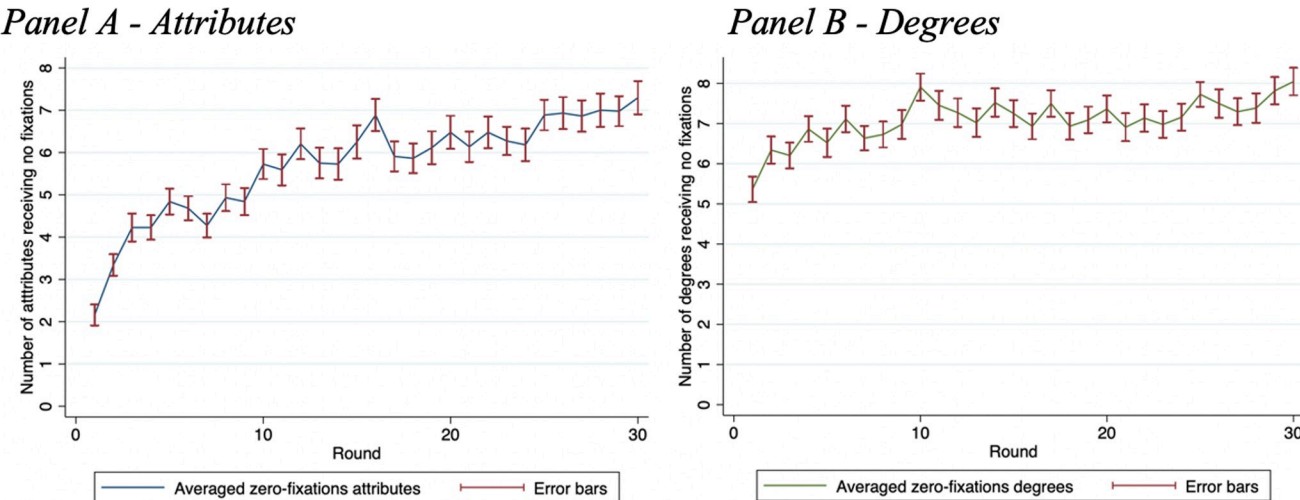

**Fig 2. Zero-fixations attributes and degrees.** Distribution of zero-fixations attributes over the succeeding scenarios for attributes (Panel A) and degrees (Panel B). The error bars are depicted in dark red.

that are viewed (based on the fixation measure) without examining their degree is 65,3%, while the number of degrees sampled without viewing their attribute is 34,7%. When studying these measures over order, we find that, while the average sum of attributes being seen without checking the respective degree is decreasing over order, the opposite trend is observed for the degrees viewed in isolation. It suggests that participants tend to focus less and less on reading the attributes over time, and more and more on noticing the level of the degrees.

We again exploit the richness of our available data by pulling together eye-tracking, PC, and Ranked data. We test the average number of zero-fixation attributes over attributes being significant or not in the individual regressions on the PC data and over the attributes ranked as the most important or not in the Ranked data. For attributes, we find a significantly lower proportion of zero fixations on attributes which are found as significant in the PC data ($t = 1.803$; $p = 0.036$) and on the attributes found to be the highest ranked ($t = 2.791$; $p = 0.002$). For degrees, the association is even more pronounced (PC data: $t = 2.125$; $p = 0.017$; Ranked data: $t = 2.660$; $p = 0.004$). Our results suggest that a clear selection occurs. We find that respondents become increasingly more selective with repetition of the PC experiment, and that those attributes getting attention are associated with the attributes identified as the most important to the respondents.

To also provide evidence on what participants in fact attend to, and not only what they do not attend to, we have performed a parallel analysis of actual fixations on attributes and degrees (each attribute and the corresponding degree is treated as one unit) for each scenario. We find that the numbers of attributes and degrees receiving fixations is significantly lower for the last ten scenarios compared to the first ten (t-test: $t = 5.34$; $p < 0.0001$). Furthermore, the middle ten scenarios are also significantly different from the first ten scenarios ($t = 3.45$; $p = 0.0003$), but not different from the last ten scenarios ($t = 0.83$; $p = 0.2022$), suggesting that the selection process main takes place in the beginning. We repeated the analysis at an individual level by comparing the individual number of attributes and degrees together fixated at in the first ten scenarios with the same person's number of fixations in the last ten scenarios. At the individual level, we confirm that the number of attributes and degrees together fixated at is significantly decreasing (paired t-test $t = 3.55$; $p = 0.0005$). Together these results underline that participants go through a process of selecting what items to attend to over the course of the study.

## Consistency (H1.3)

We operationalize consistency by obtaining a measure of violations, defined as the number of times respondents violate what they state their decisions are based on and what they actually base their decision on [49]. We calculate violations by matching the PC data and the Ranked data. More specifically, a violation occurs when an attribute found to be significant in the PC assessment regression is given a low rating in the ranking task. To quantify the consistency, the significance level of each attribute is tested against its final ranking.

We find that the average difference between the importance originating from the significance levels of the policy-capturing data and the importance from the rated data is not different from zero ($p = 1.000$), and in a Wilcoxon signed rank test we do not find a significant difference between the two rankings ($z = 0.035$; $p = 0.971$), suggesting that the two preferences measures are generally aligned. We do want to stress that substantial variation exists. The relationship between the two measures is also illustrated in Fig 3. For each of the sixteen positions of Ranked data, we illustrate the average of the associated ranking in the PC ranking data. We observe that the importance of the two measures tend to follow each other, particularly for the attributes rated as most important.

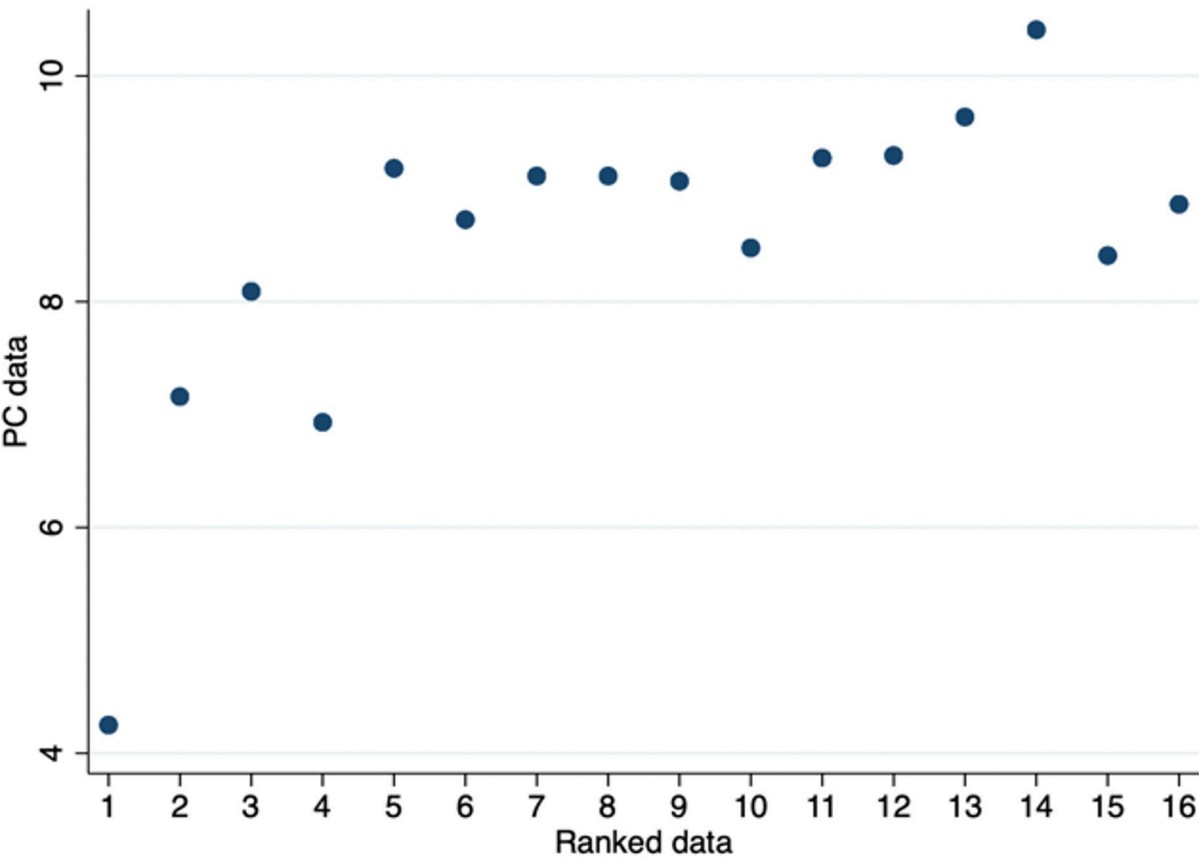

**Fig 3. Policy-capturing vs. ranked data.** The relationship between Policy-capturing data and Ranked data.

We also calculate the magnitude of violations between the two rankings [49]. In an ANOVA test, we find that the absolute magnitude of violations is not significantly different across the rankings (F = 1.510; p = 0.094).

Our data also allow us to compared consistency between the PC data and the Ranked data with the eye-tracked data (Attributes and degrees are ranked according to the average time spent on them). We find that the ranking resulting from the PC data does not deviate significantly from the ranking resulting from the eye-track data (attributes: t = -0.085, p = 0.932; degrees: t = 0.103; p = 0.917). Furthermore, the ranking in the Ranked data does not significantly deviate from the ranking resulting from the eye-tracked data (attributes: t = 0.151, p = 0.880; degrees: t = -0.023, p = 0.981). Our results thus suggest that respondents show a great degree of consistency across the PC, Ranked, and eye-tracked data, when ranking what is important to them in the collaboration opportunities.

## Experience (H2)

Experience is expected to play a role in assessing the collaboration opportunities. We study experience (years of professional experience) by separating the findings above across our subsamples of students and professionals. Overall, students spend 23.98 seconds looking at the average scenario, while professionals spend 22.88 seconds, the difference is not statistically significant (p = 0.218). Students and professionals focus their attention on different AOIs: while professionals allocate more attention to the attribute degrees than students (318 vs. 283

milliseconds per degree—p = 0.069), students spend more time reading the attribute texts (896 ms vs. 1027 ms per attribute—p = 0.048). The observed difference might be the result of working experience making it faster for professionals to understand the attributes.

## Experience and efficiency

Efficiency, the decrease of effort over time, is illustrated for the two sub-samples in Fig 4. The graph shows how the attention paid to the attributes drops dramatically from the first to the last scenario, specifically for professionals. For the degrees, the efficiency is more stable and similar for the two samples. For reference, find the graphic patterns in Fig 1.

We find that both types of respondents show a significant decrease of *time spent* on attributes, between the first ten scenarios and the last ten scenarios (p(professionals) < 0.002; p (students) < 0.001). For degrees, the decrease of time spent is only significant for students (p (professionals) < 0.125; p(students) < 0.016).

As a robustness check, we re-run the random effects regression of the logarithmic function of time spent as the dependent variable for the attributes and the degrees, respectively (Table 3).

The explanatory variable *round* is included (a range of values from 1 to 30 according to the individual chronological showcasing of the scenarios), together with the *student* dummy and the interaction between the two (*student* X *round*). The interaction term is significant for both

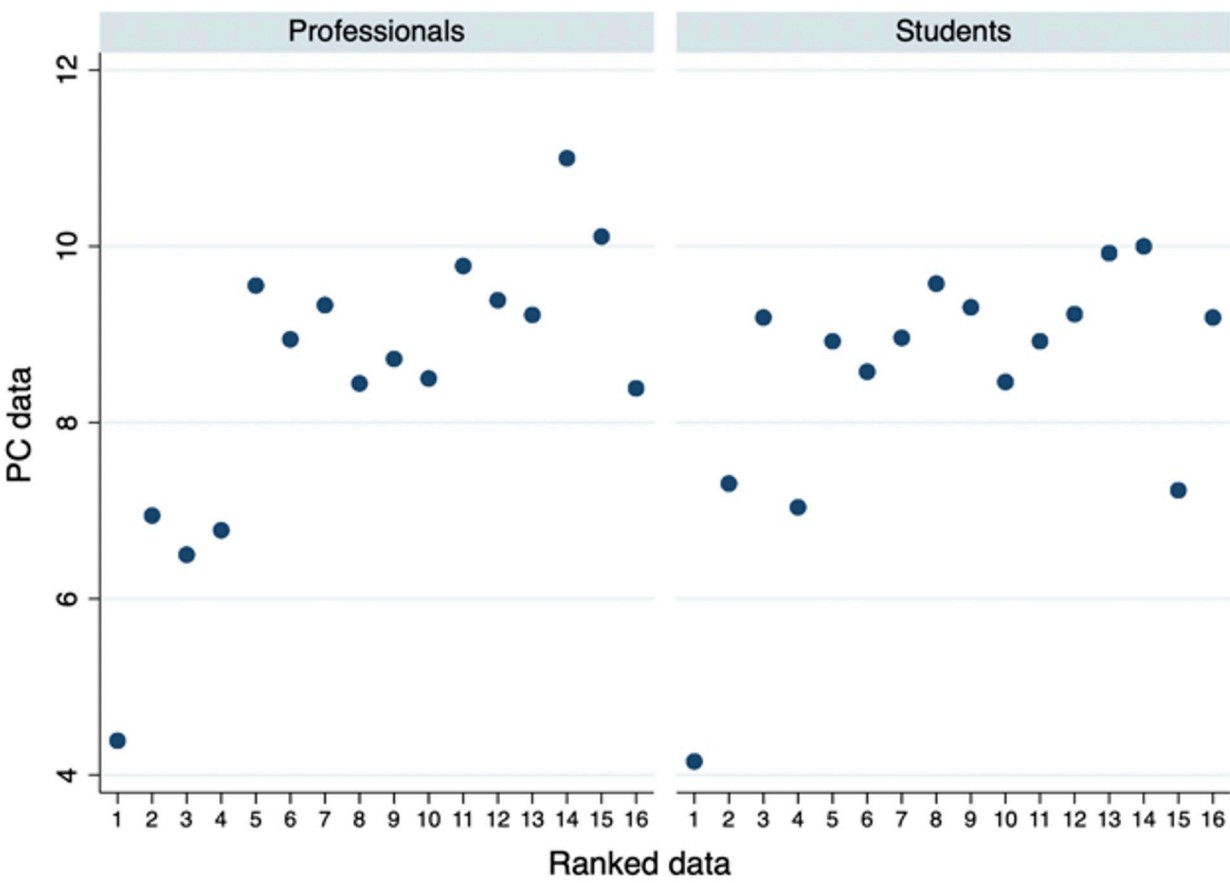

**Fig 4. Policy-capturing vs. ranked data by sample.** The relationship between Policy-capturing data and Ranked data by sample group.

**Table 3. Regression table H2.** Random effects regression models for both attributes and degrees testing hypothesis 2 (p-values within brackets).

| | Hypothesis 2—efficiency | | | | | | Hypothesis 2—selectivity | | | | | |
|---|---|---|---|---|---|---|---|---|---|---|---|---|
| | for attributes | | | for degrees | | | for attributes | | | for degrees | | |
| Rounds | -0.064*** | -0.050*** | -0.050*** | -0.022*** | -0.004 | -0.004 | 0.122*** | 0.107* | 0.107* | 0.042* | -0.002 | -0.002 |
| | (0.000) | (0.000) | (0.000) | (0.000) | (0.562) | (0.562) | (0.000) | (0.017) | (0.017) | (0.030) | (0.951) | (0.951) |
| Student dummy | | 0.362 | 0.255 | | 0.490 | 0.505 | | -0.317 | 1.143 | | -1.513 | 0.812 |
| | | (0.577) | (0.841) | | (0.441) | (0.680) | | (0.774) | (0.510) | | (0.227) | (0.648) |
| Student X Rounds | | -0.023* | -0.023* | | -0.031*** | -0.031*** | | 0.026 | 0.026 | | 0.075 | 0.075 |
| | | (0.029) | (0.029) | | (0.000) | (0.000) | | (0.607) | (0.608) | | (0.063) | (0.064) |
| Gender | | | -1.093 | | | -1.441* | | | 1.551 | | | 1.559 |
| | | | (0.146) | | | (0.048) | | | (0.248) | | | (0.245) |
| National | | | 0.627 | | | 0.783 | | | -1.216 | | | -1.475 |
| | | | (0.374) | | | (0.250) | | | (0.345) | | | (0.220) |
| Education | | | 0.042 | | | 0.030 | | | -0.027 | | | -0.060 |
| | | | (0.697) | | | (0.772) | | | (0.874) | | | (0.721) |
| Work experience | | | 0.241 | | | 0.211 | | | -0.623 | | | -0.684 |
| | | | (0.448) | | | (0.494) | | | (0.293) | | | (0.227) |
| Constant | 9.499*** | 9.285*** | 9.918** | 7.800*** | 7.510*** | 8.342* | 3.804*** | 3.991*** | 2.023 | 6.443*** | 7.337*** | 3.064 |
| | (0.000) | (0.000) | (0.004) | (0.000) | (0.000) | (0.012) | (0.000) | (0.000) | (0.736) | (0.000) | (0.000) | (0.623) |
| R-squared | 0.044 | 0.045 | 0.089 | 0.006 | 0.010 | 0.091 | 0.047 | 0.048 | 0.102 | 0.006 | 0.013 | 0.101 |
| N | 1320 | 1320 | 1320 | 1320 | 1320 | 1320 | 1320 | 1320 | 1320 | 1320 | 1320 | 1320 |

* p<0.05,

** p<0.01,

*** p<0.001

d.v. log(time spent per scenario)

d.v. sum of no fixation attributes per scenario

attributes and degrees. The outcomes do not change when controlling for a number of demographics *(years of work experience, age, gender, nationality)*. The results suggest that student tend to have a faster development in efficiency for attributes, and students tend to have increase in efficiency in their attention on degrees, which the professionals do not.

An additional interesting feature to test in the context of efficiency is the length of the wording of each attribute description. Thus, a variable indicating the number of words that each attribute is described with (e.g., the first attribute corresponding to 14, the second one with 18, etc.) was created. To investigate the role of the order of scenarios and the attributes length simultaneously, we have run an additional Random Effects regression, which replicates and adds on Table 2, in S1 Table. S1 Table shows that respondents spend systematically less time on the lengthy items over time. When splitting the sample into industry scientists and students, we notice that the effect is driven by students.

Finally, we bring together the eye-tracked data with the PC data. When testing the difference in time spent between the first ten scenarios and the last ten on attributes, no significant association is found with the significant attributes from the PC data ($p = 0.147$ for professionals; $p = 0.265$ for students). The same result applies for the comparison with the Ranked data ($t = 1.172$, $p = 0.121$ for professionals; $t = -1.370$, $p = 0.085$ for students). For the degrees, we find a significant difference in attention for the higher ranked attributes for both students ($t = 1.764$; $p = 0.041$) and professionals ($t = -1.736$; $p = 0.039$), despite the direction of the effect being negative for professionals (i.e., their attention decreases at later scenarios on the degrees belonging to highly ranked attributes).

## Experience and selectivity

An additional consequence of experience is that it boosts encoding and retrieving of information [71]. We are therefore interested in checking whether professionals focus their fixations differently areas compared to students. We replicate the analysis of Hypothesis H1.2 with a split sample between Professionals and Students. We find that both samples have a significant increase in attributes not fixated on from the first ten to the last ten scenarios, although the increase is bigger for students (professionals: difference 2.04, p = 0.014; students: difference 2.60, p<0.001). For degrees, professionals experience a small non-significant reduction in non-fixated attributes, students show however a significant increase in no-fixations (professionals: difference -0.22, p = 0.379; students: difference 1.38, p = 0.001).

We replicate the random effects regression of Hypothesis H1.2, but this time we include the *student* dummy and the interaction term (*student* X *round*), in addition to the explanatory variable *round* and the *attributes* controls. See Table 3. Both models with and without controls result in a non-significant interaction effect for either attributes or degree, suggest that students and professionals undergo a similar selection process. We also detect an association between the areas that they evaluate as important in the PC exercise and a lower number of no-fixations attributes, which is only significant for students (attributes: t = 2.108, p = 0.017; degrees: t = 2.943, p = 0.001).

## Experience and consistency

Lastly, we look at the consistency difference among the two sample groups, by replicating the same tests on the difference between PC data and Ranked data and on violations. Fig 4 plots the relationship between the ranking provided in the Ranked data exercise and the ranking resulted from the regression of the Policy-capturing evaluation. It is split by sample composition, and it shows that no clear difference in the pattern of this consistency measure emerges among professionals and students.

Surprisingly, we find no significant differences in the Wilcoxon signed rank sum test between rank data and PC data for either of the sample group (professionals: p = 0.851; students: p = 0.917), and no significant difference among the distribution of violations among attributes from the ANOVA test (professionals: F = 1.02, p = 0.438; students: F = 1.13, p = 0.322). The same result applies when we test attention evidence: no significant difference for attributes in PC data ranking vs. eye-tracked ranking (professionals: z = 0.022, p = 0.982; students: z = -0.133, p = 0.894) or in the Ranked data vs. eye-tracked ranking (professionals: z = 0.136, p = 0.892; students: z = 0.078, p = 0.938); and no significant difference for degrees in PC data ranking vs. eye-tracked ranking (professionals: z = 0.191, p = 0.848; students: z = -0.043, p = 0.966) or in the rank data vs. eye-tracked ranking (professionals: z = 0.082, p = 0.935; students: z = -0.130, p = 0.896). Both students and professional seem to be consistent in their rankings of what is important to them in the collaboration opportunities.

Another internal measure of consistency is the extent to which each respondent's choice is consistent with their judgement by the means of the R-squared of their individual regressions. To address this, we average the individual R-squared of the first half of scenarios and the individual R-squared of the second half to see how much it varies. The R-squared does not show an increasing trend. Moreover, when testing the values of the R-squared for the first ten scenarios against the last ten scenarios on the overall sample, no significant effect is detected (p = 0.507). When splitting the sample by students and industry scientists, no significant effect is found either, which suggest that experience seems not to have an impact on the levels of R-squared of the individual regressions. This result suggests that the individual consistency does not develop over the time course of the experiment, but it remains stable throughout.

A final step of our analysis is to run two regressions with time spent jointly on the attribute and the corresponding degree as dependent variable. The regressions are listed in Table 4. In the first regression, model 1, we apply attribute length, stated disliking (the ranked data of attributes from most liked (= 1) to least liked (= 16)) and order as explanatory variables. In the second regression, model 2, we furhermore add two-way interaction effects between the explanatory variables and a three-way interaction between all of them. Both regressions control for individual dummies and the attribute dummies. All explanation variables are normalized. The first regression highlights that more time is generally spent on longer attributes (everything else kept equal), and that less time is spent on less liked attributes, while time spent is reduced over the course of the policy capturing experiment. In addition, the second regression shows that the longer time spent on more lengthy attributes is fading out over the repetition of the scenarios, suggesting that less time is spent reading the actual content of the attributes over time, but rather the time is spent observing what degrees the different attributes are scaled at.

## Discussion & conclusion

In this study, we seek to investigate the cognitive processes behind the policy-capturing technique, as measured by eye-tracking. Our objective is to study the information acquisition and cognitive processes that respondents undergo as they review a series of scenarios while participating in a PC experiment. We do so by analyzing what characterize the cognitive mechanisms that arise as our respondents assess the attractiveness of 30 randomly ordered scenarios.

We find firstly that respondents become more efficient with practice in the PC experiment. The effort required to accrue information about the collaboration scenarios is decreasing during the PC exercise.

**Table 4. Triple interaction regression table.** Regression models of the relative impact of attribute length, individual ranking, and scenarios order on the time spent on attributes and corresponding degrees (p-values within parentheses). The three explanatory variables are normalized.

| | Model 1 | Model 2 |
|---|---|---|
| Attribute length | 105.137** | 106.130** |
| | (0.009) | (0.009) |
| Stated disliking (Ranked Data) | -69.628*** | -69.710*** |
| | (0.000) | (0.000) |
| Scenario order | -387.903*** | -388.844*** |
| | (0.000) | (0.000) |
| Attribute length X Stated disliking (Ranked Data) | | -19.554 |
| | | (0.130) |
| Attribute length X Order | | -42.331*** |
| | | (0.000) |
| Stated disliking (Ranked Data) X Scenario Order | | 4.773 |
| | | (0.693) |
| Triple interaction | | 23.511 |
| | | (0.057) |
| Individual dummies | Yes | Yes |
| Attribute dummies | Yes | Yes |
| Constant | 1348.483*** | 1349.410*** |
| | (0.000) | (0.000) |
| R-squared | 0.26 | 0.26 |
| N | 21120 | 21120 |

Secondly, we find that respondents are selective in their information acquisition as they undergo an increasing number of non-fixated information over the course of the experiment. Moreover, we find that they direct attention towards the attributes that appear to be more important to them. Therefore, a clear selection pattern among the important attributes that characterize the scenarios emerges over time.

Thirdly, we find a consistent link between what respondents evaluate as important attributes in the PC and what they state is important for their evaluation.

Fourthly, we find that the cognitive processing of answering the PC experiment is surprisingly similar among students and professionals. While detecting that students are quicker in reducing their cognitive effort, for instance, we do not observe systematic differences in the amount of time allocated, the pattern of attention and the selective information acquisition process among the split samples. There are some tendencies that underline small differences among subjects but, overall, our findings seem to suggest that students may be a good proxy for more experienced decision makers.

Our results confirm that respondents in policy-capturing studies develop mental shortcuts for how to handle the massive amount of information intentionally provided. As such, our findings support a common assertion of many policy-capturing studies that assume participants develop policies as they review a series of scenarios while participating in a PC experiment, and that these policies influence their information processing and the judgements they make. Our findings show little difference among experienced and less experienced respondent. It is tempting to make the immediate conclusion that the convenient sample, students, can just as well be used as respondents in a future PC student, but one central point to notice is that we find similar processes and consistency among students and professionals, which is not the same as them giving the same evaluations and the same emphasis on certain pieces of information. The outcome of experienced respondents' evaluation might still be different. All we are concluding is that the process and the consistency are similar across students and professionals.

Policy-capturing experiments share many similarities with studies in choice modelling, and our study draw on insights from research at the intersection of choice-modelling and eye-tracking, when building its hypotheses [48, 49]. It is interesting to note that despite the similarities with policy-capturing experiments, our study produces somewhat different outcomes. Where we find, through attention, that our participants develop cognitive models to make their decisions, evidence from discrete-choice experiments and eye-tracking combined does not find documentation of similar cognitive model being developed, but rather that participants follow systematic search patterns in their information acquisition [52, 116]. A potential reason for this difference could be that policy-capturing studies include more extensive amounts of information in each scenario to be considered compared to choice modelling studies, resulting in a larger need for developing cognitive models to cope with the situation. Another noteworthy difference relates to areas or attributes of a choice situation that are not attended to: attribute non-attendance (ANA). Whereas the eye-tracking discrete-choice experiments find mixed results about how visual ANA, stated ANA, and inferred ANA relate [83, 117] we find that our participants are generally able to state and attend to the attributes they put weight on in their policy assessments. Despite the differences that do exist across policy-capturing and discrete choice experiments, we believe that not only does our study benefit from bridging evidence from eye-tracking discrete-choice modelling research, but that such approach should represent a general inspiration for future studies to integrate insights from related disciplines.

Our findings can be viewed as an application of the eye-mind hypothesis. Our participants tend to attend more to the elements which are both revealed to be important for their answers

to the policy-capturing exercise but also rated by themselves as important, suggesting that the eye-mind hypothesis fits also with our applied setting.

Finally, a fundamental methodological contribution of our study is to showcase the potential gains of combining an existing empirical methodology with attention evidence. Such combination allows researchers to go beyond the assumption of a particular cognitive process and to explicitly map the process instead. Methods, such as vignette studies, are often built on more or less explicit cognitive assumptions. We show that a promising way to conduct future research is to directly investigate the information acquisition and cognitive processes of such instruments through attention measures.

Our study is the first to combine eye-tracking and policy-capturing. Although providing novel insights on the attention process during a policy-capturing study, a number of limitations naturally applies. First, following the standard choice design in policy-capturing methods, we keep the order of the displayed items fixed. It would be interesting to randomly vary the order to create a causal relation on micro processes such as center bias and top-down processes. Second, we fixate the information amount of each scenario, following the standard method specification. Randomly varying the amount of information would allow to determine whether using more information indeed results in participants creating heuristics, as assumed in policy-capturing studies.

We hope future studies will appreciate our initial findings that participants do build cognitive shortcuts to complete the policy-capturing exercise, but also continue to explore more how and when those shortcuts develop.

## Supporting information

**S1 Fig. Scenario layout.** An example of the structure of one of the 30 scenarios and the Areas of Interest distribution.
(TIF)

**S1 Table. Student-driven effect.** Random Effects regression showing respondents' systematic decrease in attention on the lengthy items over time.
(TIF)

## Author Contributions

**Conceptualization:** Alice Pizzo, Toke R. Fosgaard.

**Data curation:** Alice Pizzo.

**Formal analysis:** Alice Pizzo.

**Funding acquisition:** Toke R. Fosgaard, Beverly B. Tyler, Karin Beukel.

**Investigation:** Alice Pizzo.

**Methodology:** Alice Pizzo, Toke R. Fosgaard, Beverly B. Tyler.

**Project administration:** Toke R. Fosgaard, Beverly B. Tyler, Karin Beukel.

**Resources:** Toke R. Fosgaard, Beverly B. Tyler, Karin Beukel.

**Software:** Alice Pizzo.

**Supervision:** Toke R. Fosgaard.

**Writing – original draft:** Alice Pizzo.

**Writing – review & editing:** Alice Pizzo, Toke R. Fosgaard.

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
