## [Decision Letter · Decision Letter 0]

8 Apr 2022

PONE-D-22-03769Information acquisition and cognitive processes during strategic decision-making: combining a policy-capturing study with eye-tracking data.PLOS ONE

Dear Dr. Pizzo,

Thank you for submitting your manuscript to PLOS ONE. After careful consideration, we feel that it has merit but does not fully meet PLOS ONE’s publication criteria as it currently stands. Therefore, we invite you to submit a revised version of the manuscript that addresses the points raised during the review process.

I have received feedback on your manuscript from two expert reviewers. Both reviewers and myself see merit in your work. However, as the manuscript currently stands, it presents important shortcomings that must be carefully addressed before it could be publised. The issues include data availability, data analysis, connection to the literature and to previous results, and many other. Hence, you should consider this revision opportunity a high risk endavour. In case you decide to undertake the improvement task requested, you must know that I will ask the same two reviewers to consider again your paper for publication.

We look forward to receiving your revised manuscript.

Kind regards,

Iván Barreda-Tarrazona, PhD

Academic Editor

PLOS ONE

Journal Requirements:

“NO”

Reviewers' comments:

Reviewer's Responses to Questions

**Comments to the Author**

1. Is the manuscript technically sound, and do the data support the conclusions?

Reviewer #1: Partly

Reviewer #2: Partly

2. Has the statistical analysis been performed appropriately and rigorously? 

Reviewer #1: No

Reviewer #2: Yes

3. Have the authors made all data underlying the findings in their manuscript fully available?

Reviewer #1: No

Reviewer #2: No

4. Is the manuscript presented in an intelligible fashion and written in standard English?

Reviewer #1: Yes

Reviewer #2: Yes

5. Review Comments to the Author

Reviewer #1: This paper provides an interesting use of eye tracking to uncover the use of information in a policy capturing context. Unfortunately the results you found are relatively uninteresting and only marginally useful to improve policy capture. Here are a number of suggested changes that might strengthen the paper.

1. Relate judgment micro-processes to actual valuations. Understanding the attentional processes is important if it alters the results defined in terms of the derived policy evaluations of the attributes. Currently this seems like half a paper.

2. Clarify your design. It is hard to understand the task used or the design across respondents. The order of the scenarios is randomized across respondents. Is the order of the attributes within scenarios randomized, or their attendant valuations? You indicated that correlation between the 16 attributes (defined linearly 1-5) is less =/- .40 across the 30 tasks. It is surprising you could not find a design with lower correlations. There should be a design with 32 tasks that can generate 16 independent binary variables. Your example design in Figure 1A shows no evidence of any of the attributes having a low (1) rating. Is that true of all scenarios? Having 4 levels may help you find an orthogonal design. Is that what you did?

3. Do you need both efficiency and selectivity? Conceptually, these constructs are inversely related, as a processes which are selective are likely to be less efficient. What is their correlation? The inverse similarity is evident in their statistical patterns is shown in Table 2. It might be wise to focus only on selectivity, as that is normative, provided, as you show, that one attends the attributes that are less likely to be important.

4. Play down the student vs. employed effect. The least interesting findings of this paper stem from the low difference between the processing of students vs. employed. Part of the problem may stem very small samples of respondents and from the inherent variability within categories. The only difference that matters is that it takes a little longer for the students to get used to the attributes, as one would expect. Do the two groups differ in their valuation of attributes?

5. Do a better job with accuracy. Accuracy is key. Currently you only measure it based on the extent to which attention focuses on attributes that are valued by a respondent. A second measure is the extent to which each respondent’s choice is consistent with their judgments, assessed by either the standard error or the R-squares of within-individual regressions. Typically, that internal measure of consistency increases with round as respondents adjust their responses to be consistent with each other. A third measure of consistency is whether a respondents’ direct measures differ with that of their peers. It would be exciting to show that peer consistency increases with round, implying that they merge towards each other’s judgments over time.

6. Separate aggregate from individual analysis. It would be helpful to display a regression pooling all respondents together. That enables tests of shared strategies across individual. Then run regressions within individuals. Those individual tests can assess the extent to which individuals used different strategies. Below are four potentially valuable tests.

7. Test the impact of attributes near the top. Check to see whether most searches move from top to bottom. If that occurs, there may be enough information for a reasonable evaluation to be made for items early in the process, thus limiting time spent at a low accuracy cost.

8. Test the impact of the lengths of the attribute descriptions. With experience, the time taken focused on an attribute may drop because respondents can recognize its value without having to reread the sentence. This drop is more likely for long descriptions.

9. Test whether some respondents may be ignoring the attribute labels and simply averaging degree information. That shortcut is even easier if one evaluates the scenario by its number of 5’s. More complex but almost as easy is the number of 4’s or 5’s, or the number of 5’s minus the number of 1’s. It should be possible to measure saccades down the column to see if they are scanning down to look for other similar ratings.

Finally, test whether respondents are first exploring important attributes and then examining their ratings. A number of respondents initially examine the attributes top down and then examine the degree for each. If so, you will find attribute-to-degree moves but few in reverse. The selectivity then comes in by moving immediately to a few important attributes.

Reviewer #2: The authors present an analysis of eye movements and decisions during a policy capturing (PC) experiment in which participants rated the potential of 30 possible academic collaboration opportunities. Looking at the literature, this seems to be the first, or one of the first studies to use eye tracking to study decision strategies in a policy capturing paradigm. The paradigm, however, resembles the discrete choice experiment (DCE) to quite an extent and in the context of this paradigm, eye tracking has been used extensively. The authors refer to the DCE eye tracking papers, but I think better use could be made between the overlap, and a clearer discussion of the differences as well. Because this is (one of) the first paper(s) studying eye movements in PC, I think the paper makes a significant contribution. That said, I would have several comments that I think need addressing before I can recommend publication.

(1) The authors indicate that they cannot share data due to sensitive content. This is somewhat surprising, because the combination of expert/novice, fixation information, and decisions for the various policies does not allow for tracing the data back to participants. I therefore think the data should be made available (using participant numbers). With these data, I think the exact contents of the presented policies should also be made available. Any demographics such as age and gender can be omitted from the shared data, as they are not used for the analysis and could lead to identification of the participants.

(2) Various plots are shown as bar plots. I think it would be better to use line plots here, using error bars to indicate the variability across participants.

(3) As indicated, the policy capturing (PC) paradigm seems to be quite similar to discrete choice experiments (DCEs), for which there is an extensive literature on whether or not eye movements may reveal cognitive processing. I think you could do more with this literature in your introduction, for example, to make predictions, and in your discussion, to explain why the data are the way that they are. For example, the conclusion from the DCE literature seems to be that participants follow a fairly standard trajectory through the information presented on the screen (very much top-to-bottom, left-to-right), but that eye movements provide relatively little information about cognitive processing (which would be in line with your observations that experts and novices show similar eye movement patterns). Likewise, an important topic in the eye tracking DCE literature is attribute non-attendance (ANA), where three ways of measuring ANA have been identified: stated ANA, inferred ANA and visual ANA (the latter on the basis of eye movements). Studies have suggested that these three may not always be aligned. This could be in contrast with your finding that participants are well able to indicate which attributes are important for their decisions.

(4) That said the policy capture method seems to differ from the DCE method in that many more attributes are shown (typically DCEs have around 4 to 7 attributes), that ratings are collected (rather than choices between options), and that no systematic method seems to be applied to decide which attributes and which levels to present to participants (in DCEs a method called d-efficient designs are used to optimally select attribute levels across choice tasks). I think it would be important to contrast the two methods in your work, and explain what can be learned from the DCE literature for PC and what cannot.

(5) I think the discussion could be more like a discussion. I would very much like to see a comparison between the present results and past findings (there is quite a bit of literature on eye movements for discrete choice experiments), discussions of any differences with past findings, how results fit in existing theories (such as the eye-mind hypothesis mentioned in the introduction) and a discussion of possible limitations of the study and future directions.

(6) I found it difficult to understand how the eye movement data were recorded and analysed. The survey seems to be presented in Qualtrics, which seems to be an online survey tool. Eye movements seem to be collected with a Tobii T60 eye tracker. How was the onset of each question aligned with the eye movements?

(7) For the fixations, did you make use of the automatic segmentation method of the T60 eye track to separate samples into fixations and saccades?

(8) I’m a bit worried about line 208 where you indicate to only use fixations of at least 200ms. That is quite a high threshold. I have seen thresholds of 80ms being used, but looking at the fixation distributions provided by Rayner in his articles and books, I think you may be missing quite a substantial number of fixations in your analysis by using a 200ms cut-off duration.

(9) In line 255 you indicate to follow the method by Hitt et al and Tyler et al. I think it would be important to briefly explain what these methods involve.

(10) Part of the participants seem to have participated online (no eye tracking) and part took part in the eye tracker. What is unclear to me is how these data were analysed. Were eye tracking outcomes based on just the eye movement participants while the decision data were based on all participants? What was done for analyses that involved the combination of decisions and eye tracking data?

(11) The time spent (often called dwell times) seem to be reported in seconds. This may be problematic, because some participants take a long time to decide, while others take less (so those who spend more time on each task contribute more strongly to the average data). Time spent on tasks also decreases over time, and therefore tasks early in the sequence contribute more than later tasks. I would strongly recommend to also consider time spent as a percentage of the total trial duration, to reduce such length effects.

(12) It is unclear to me what the purpose of the pilot was. How much was changed to the study protocol after the pilot?

(13) Please make sure that all statistics are reported with the same number of digits (sometimes there are too many).

(14) Please have the paper proof-read one further time before resubmiting.

6. PLOS authors have the option to publish the peer review history of their article (what does this mean?). If published, this will include your full peer review and any attached files.

Reviewer #1: No

Reviewer #2: No

---

## [Author Response · Author response to Decision Letter 0]

13 Jun 2022

Thank you for taking into consideration our manuscript further. 

We have made a careful effort to address all issues raised by the reviewers and we have included most comments to the manuscript, which now stands as an improved and more complete version. We have clarified the text where needed, and expanded the reference and comparison to the literature and previous results. We have made the data available, once anonymized. Finally, new results were pulled out from the data as suggested by the reviewers or in order to address specific comments. The majority were also integrated in the manuscript.

All details of our revisions are included in the Response to Reviewers document.

We remain available for any inquiry you may have.

---

## [Decision Letter · Decision Letter 1]

25 Aug 2022

PONE-D-22-03769R1Information acquisition and cognitive processes during strategic decision-making: combining a policy-capturing study with eye-tracking data.PLOS ONE

Dear Dr. Pizzo,

Thank you for submitting your manuscript to PLOS ONE. After careful consideration, we feel that it has merit but does not fully meet PLOS ONE’s publication criteria as it currently stands. Therefore, we invite you to submit a revised version of the manuscript that addresses the points raised during the review process.Both reviewers see great progress in the way you have responded to their comments. In fact reviewer 2 just wants you to provide an easier to interpret dataset, so I ask you to include a file with the description of each of the variables in the dataset. Please also make sure to carefully consider all suggestions that reviewer 1 is putting forward. After the next round of revision I will make a final decision about the publishability of the paper.

We look forward to receiving your revised manuscript.

Kind regards,

Iván Barreda-Tarrazona, PhD

Academic Editor

PLOS ONE

Reviewers' comments:

Reviewer's Responses to Questions

**Comments to the Author**

1. If the authors have adequately addressed your comments raised in a previous round of review and you feel that this manuscript is now acceptable for publication, you may indicate that here to bypass the “Comments to the Author” section, enter your conflict of interest statement in the “Confidential to Editor” section, and submit your "Accept" recommendation.

Reviewer #1: All comments have been addressed

Reviewer #2: All comments have been addressed

2. Is the manuscript technically sound, and do the data support the conclusions?

Reviewer #1: Partly

Reviewer #2: Yes

3. Has the statistical analysis been performed appropriately and rigorously? 

Reviewer #1: I Don't Know

Reviewer #2: Yes

4. Have the authors made all data underlying the findings in their manuscript fully available?

Reviewer #1: Yes

Reviewer #2: No

5. Is the manuscript presented in an intelligible fashion and written in standard English?

Reviewer #1: No

Reviewer #2: Yes

6. Review Comments to the Author

Reviewer #1: (No Response)

Reviewer #2: My apologies for the delay in my response. It is quite a long paper and it has been a few difficult months with everyone else "going back to normal". Thank you for offering the version that shows the changes and the detailed letter with the responses to the comments. This has helped a lot. I think all my comments have been addressed in the revision. When reading the revision I thought that the introduction could have had a bit more information on the similarities and differences between the policy capturing paradigm and discrete choice experiments, but I think the current layout with this information in the discussion also works. I am glad to see that the cut-off for fixations is 100ms (which is a common threshold), instead of 200ms. I also looked through the replies to the other reviewer, and very much appreciate that you back up your responses with relevant analyses.

The one remaining request I would still have, is to provide an overview of what each of the variables in your data-set mean (best shared along with the data-set). I had to look up the .dta format and found that it is from STATA, which I can read in R using the "haven" package, so this works out well (but you may want to add that information as well). For example, what is contained in the variables: qc1-qp1-qp2-qp3-qp4-qp5-qp6-qp7? What is in c1-c2-c3-c4-c5-c6-c7-c8-c9-c10-c11-c12-c13-c14-c15-c16-d1-d2-d3-d4-d5-d6-d7-d8-d9-d10-d11-d12-d13-d14-d15-d16-a1-a2? Some guidance on what is what would be helpful.

7. PLOS authors have the option to publish the peer review history of their article (what does this mean?). If published, this will include your full peer review and any attached files.

Reviewer #1: **Yes: **Joel Huber

Reviewer #2: No

---

## [Author Response · Author response to Decision Letter 1]

2 Oct 2022

Note: We report here below the response to reviewer. It is a copy of the document attached to this revision. We suggest to read it from the doc file. Figures here below are missing.

Dear Editor, Dear Reviewers,

Thanks for the opportunity to resubmit our paper. We are very thankful for your constructive comments and suggestions which are all well-taken. Below, we are responding to all the comments and are highlighting the changes made in the manuscript as a reaction to the comments (our answers are in blue). Overall, the comments and the suggested changes have, in our opinion, improved the manuscript substantially. We would like to mention that we found a few mistakes in the first “response to reviewers” document. For that reason, we are re-submitting the first reply document (with the corrections), along with this second reply document. 

Reviewer 1:

Thank you for your careful responses to the questions in the first draft. I believe there are some important issues that need to be resolved for this to be an important paper that helps understand multi-attribute scenario judgments.

Initial Reactions: 

1. Figures 1 and 4. Figure 1 shows time by task order. Perhaps convert both to log scale. Double log learning curves tend to be quite linear. Figure 4 provides very little information not found in Figure 1. You might simply present Figure 4 first and then table 4 showing differences in time spent differ little by professionals and students. 

Thank you for the suggestion. We have followed your suggestion by converting the Average Time Spent of the following exhibit into the logarithmic form. In the Manuscript, we keep Figure 4, now renamed as Figure 1, and we take away the old version of Figure 1.

The old version of Figure 4 is dropped, Figures 5 and 6 are now Figure 4 and Figure 5 (see additional change in answer 3). Figure 1 includes the distinction across the professional and the student sample, and the text was updated as follows:

“The development of attention over time is shown in Fig 1, which also includes the split between Professionals and Students.”

Fig 1. Attention. The pattern of attention; time spent (seconds), log transformed, for attributes (Panel A) and degrees (Panel B) over the 30 repeated collaboration scenarios, by professionals and students.

2. Figure 2. Attributes-not-viewed. Are these relevant differences? It might be more meaningful to provide the % of information viewed per attribute rather than a binary scale of the proportion of attributes viewed vs. not viewed. 

Thanks for the comments, which are well-taken. Unfortunately, we do not have access to the original raw data from the study due to a change of institutional affiliation. For this reason, we cannot reshape our Areas of Interest (AOIs) and consequently we cannot retrieve data on the percentage of information viewed per attribute (i.e., measure how many words of each item have been viewed). We would also be concerned with the data reliability as the precision the applied eye-tracker is not high enough to create AOIs for each word. 

We like the suggestion though, and have created alternative measures, displayed below, which are showing how many average fixations our participants have across the attributes and the degrees.

Figure 2-bis. Fixations. Additional Panels on the average number of fixations of all attributes per each scenario for both degrees and attributes. 

The figure illustrates the average intensity of fixation per scenario actually looked at. The trend in both panels resembles, as one can expect, the course of the time spent on each scenario, as addressed in the first hypothesis. We make a comment on this in the manuscript now. It reads as follows:

“As a contrast to the zero-fixations, we also analysis what is in fact fixated on. A paired t-test comparing the average fixations of the first ten scenarios with average fixations in the last ten scenarios finds a significant difference for attributes (p<0.0001), with 28.8 fewer fixations on average. For the degrees, the difference is also significant (p= 0.0018), with 3.1 fewer average fixations.”

3. Figure 5 shows zero-fixation attributes and degree by sample. Consider dropping Figure 5 since there are visual differences that seem apparent but are not statistically reliable. You could simply include student vs. Professional as an analysis table. 

Thanks for the suggestion. We have dropped Figure 5 and kept the discussion about the subgroup’s tests in the manuscript. The exhibit related to this result is still Table 3. Because Figure 5 was dropped, the exhibit included in the following section Experience and Consistency is now named Figure 4.

4. Figure R2 does not make sense as the three graphs are virtually identical.

There was indeed a mistake on the exhibits we had uploaded for Figure R2 Panel A and B. We are grateful you have spotted it and we have now substituted them. We find that the updated exhibit is relevant and have decide the keep it in the text. 

Fig R1.2. Accuracy. Plot of standard deviation of individual R-squared for the overall sample (Panel A) and for the split sample (Panel B).

Panel A – Overall sample Panel B – Split Sample

Suggested changes:

1. Play down the distinction between attributes and degrees. One possibility is that respondents may simply average the degrees for each scenario, but that would result in null differences between the attributes. A good normative rule specifies that the more important the attribute the more valuable it is to assess its degree. Generally, it is wasteful to spend relevant time on an attribute without considering its degree. If so, then there should be few examples where a person fixates on an important attribute without noting its scale. To test that calculate the percent of attributes that are viewed without examining its degree, and the number of degrees sampled without viewing its attribute. Those may be a small percent (say 20%) and should decrease with round or in cases where substantial time is spent on either (say > 500 ms). Below I suggest an analysis that merges attribute and degree time into one measure of attribute attention.

Thank you for the insightful suggestion. On average, the attributes that are viewed (based on fixation measure) without examining their degree is 65,3%, while the number of degrees sampled without viewing their attribute is 34,7%. The overall picture is that for 45,7% of the attributes both attributes and degrees are checked, in 25,7% neither of them are checked, while in 28,5% either attribute or degree is checked, but the corresponding degree or attribute is not checked.

When plotting this measure over order, we find that, while the average sum of attributes being seen without checking the corresponding degree is decreasing over order, the opposite trend is observed for the degrees viewed in isolation.

Fig R2.2. Information in isolation. Plots of the share of attributes viewed without viewing the corresponding degree and vice-versa.

We added this additional analysis to the manuscript, which reads:

“Another level of analysis concerns whether attributes and degrees are viewed in isolation, meaning that one is viewed while the corresponding counterpart is not. In other words, we calculate the percent of attributes that are viewed without examining their degree, and the number of degrees that are viewed without examining their attribute. On average, the attributes that are viewed (based on the fixation measure) without examining their degree is 65,3%, while the number of degrees sampled without viewing their attribute is 34,7%. When studying these measures over order, we find that, while the average sum of attributes being seen without checking the respective degree is decreasing over order, the opposite trend is observed for the degrees viewed in isolation. It suggests that participants tend to focus less and less on reading the attributes over time, and more and more on noticing the level of the degrees”

2. Play down the selectivity metric from zero fixations. The problem with eye tracking is that many fixations occur randomly or to they are briefly noted and then ignored. On Table 4 the standard errors from the selectivity measures are substantially larger than the efficiency measures. One way to fix that is to define a higher cutoff for what makes a fixation, say at 500 ms. The other way is to simply focus more on what people spend more time on rather than not. Put differently, the important measure is the total time of fixations rather than the distinction between attributes with no fixations vs. some.

Thanks a lot for the comment. We agree that it is interesting to not only look at what participants do not look at, but also what they in fact look at. We have therefore decided to add measures on the number of fixations and time spent looking at the attributes. We cannot change the fixation cut-off because we do not have access to the raw data (as mentioned above). 

Regarding the standard error (we assume it is Table 3 and not 4 which is referred to), we would like to highlight that the table is listing the p-values in the parentheses (the explanation of the parenthesis is also included in the title). 

After following the suggestion in your comment, we have added an additional analysis and updated the manuscript such that it now reads: 

“To also provide evidence on what participants in fact attend to, and not only what they do not attend to, we have performed a parallel analysis of actual fixations on attributes and degrees (each attribute and the corresponding degree is treated as one unit) for each scenario. We find that the numbers of attributes and degrees receiving fixations is significantly lower for the last ten scenarios compared to the first ten (t-test: t = 5.34; p<0.0001). Furthermore, the middle ten scenarios are also significantly different from the first ten scenarios (t = 3.45; p = 0.0003), but not different from the last ten scenarios (t = 0.83; p = 0.2022), suggesting that the selection process main takes place in the beginning. We repeated the analysis at an individual level by comparing the individual number of attributes and degrees together fixated at in the first ten scenarios with the same person’s number of fixations in the last ten scenarios. At the individual level, we confirm that the number of attributes and degrees together fixated at is significantly decreasing (paired t-test t = 3.55; p = 0.0005). Together these results underline that participants go through a process of selecting what items to attend to over the course of the study.”

3. Make a run of 21,000 observations similar to what you did in R2. The initial large estimates and small error terms in R2 lack credibility. Perhaps the measure is in milliseconds? The small errors may come from assuming that the errors within persons and tasks are uncorrelated with each other. If so, better results may come from the following analysis: 

Thank you for your precision. As for the previous comment, the digits in the parentheses of Table R2 are actually the p-values, not the standard errors. We forgot to mention this in the Table note. It is now added. The measure is in milliseconds. We address the suggested analysis in the following point.

4. Predict attribute plus degree seconds with the following independent variables: fixed constants for each of the 16 attributes, fixed constants for each of the 43 respondents, continuous measures of attribute length, individual’s ranking for each attribute, and rounds. The main effects will show the relative impact of attribute length, individual ranking, and rounds. Then interactions between rankings, attribute lengths, and rounds (zero-centered) will indicate whether rankings become more predictive of decision time with rounds, whether attribute length becomes less predictive with rounds, and whether attribute rank and length and ranking interact to predict attribute time. The idea is to identify the important factors that alter decision time.

Thanks for suggesting this analysis. It is a very useful way to understand the data. Below you may find the results of the regressions. We have also included the regressions in the manuscript as Table 4, and we have added the following text: 

“A final step of our analysis is to run two regressions with time spent jointly on the attribute and the corresponding degree as dependent variable. The regressions are listed in Table 4. In the first regression, model 1, we apply attribute length, stated disliking (the ranked data of attributes from most liked (=1) to least liked (=16)) and order as explanatory variables. In the second regression, model 2, we furhermore add two-way interaction effects between the explanatory variables and a three-way interaction between all of them. Both regressions control for individual dummies and the attribute dummies. All explanation variables are normalized. The first regression highlights that more time is generally spent on longer attributes (everything else kept equal), and that less time is spent on less liked attributes, while time spent is reduced over the course of the policy capturing experiment. In addition, the second regression shows that the longer time spent on more lengthy attributes is fading out over the repetition of the scenarios, suggesting that less time is spent reading the actual content of the attributes over time, but rather the time is spent observing what degrees the different attributes are scaled at.”

Table 4. Triple interaction regression table. Regression models of the relative impact of attribute length, individual ranking, and scenarios order on the time spent on attributes and corresponding degrees (p-values within parentheses). The three explanatory variables are normalized.

 

Reviewer 2:

My apologies for the delay in my response. It is quite a long paper and it has been a few difficult months with everyone else "going back to normal". Thank you for offering the version that shows the changes and the detailed letter with the responses to the comments. This has helped a lot. I think all my comments have been addressed in the revision. When reading the revision I thought that the introduction could have had a bit more information on the similarities and differences between the policy capturing paradigm and discrete choice experiments, but I think the current layout with this information in the discussion also works. I am glad to see that the cut-off for fixations is 100ms (which is a common threshold), instead of 200ms. I also looked through the replies to the other reviewer, and very much appreciate that you back up your responses with relevant analyses.

Thanks very much for the kind words.

The one remaining request I would still have, is to provide an overview of what each of the variables in your data-set mean (best shared along with the data-set). I had to look up the .dta format and found that it is from STATA, which I can read in R using the "haven" package, so this works out well (but you may want to add that information as well). For example, what is contained in the variables: qc1-qp1-qp2-qp3-qp4-qp5-qp6-qp7? What is in c1-c2-c3-c4-c5-c6-c7-c8-c9-c10-c11-c12-c13-c14-c15-c16-d1-d2-d3-d4-d5-d6-d7-d8-d9-d10-d11-d12-d13-d14-d15-d16-a1-a2? Some guidance on what is what would be helpful.

Thanks for your comment. We have created this detailed description of the dataset. It is printed here below, but we also attach it as a separate document to the re-submission. 

Dataset and variable description

Data Structure

For each of the 44 individuals, we have PC data for 30 scenarios.

For each scenario, we have eye-tracking data for 16 criteria, 16 degrees, 2 answers.

Therefore, the dataset is composed by 44 X 30 X 16 X 16 X 2 = 46200 observations

PC data

id = id number randomly assigned

order = ordinal variable indicating the order on which each scenario (out of 30) was shown to each subject

group = subsample dummy

scenario = descriptive name for each of the 30 scenarios

durationinseconds = how many seconds it takes to subjects to answer to the online survey

qc1 = consent page dummy (=1 if consent was provided)

age = sociodemographic numeric variable for age

gender = dummy variable for gender 

gender_long = character variable for gender

nation = dummy variable for being a local citizen or a foreigner

nationality = character variable for citizenship

edu_level = categorical variable on education level

years_work_experience = numeric variable on years of working experience following most recent education title 

years_research = numeric variable on years of working experience in research

partner_choice = dummy variable indicating whether the subject is able to choose their own collaborative partner

project_choice = dummy variable indicating whether the subject is able to choose their own collaborative project

current_collab = dummy variable indicating whether the subject is currently working on a cross-sector collaborative project 

item_1a - item_16a = categorical variable on the ranking position of each item provided by each subject in the ranking exercise of the PC experiment (ranging from 1 to 16)

item_1 - item_16 = categorical variable for degrees (ranging from 1 to 5) characterizing each criteria level

Answer = average numeric variable of Answer 1 and Answer 2

Answer1 = categorical variable describing the assessment likert scale (ranging from 1 to 7) for the first evaluation question of each scenario

Answer2 = categorical variable describing the assessment likert scale (ranging from 1 to 7) for the first evaluation question of each scenario

student = dummy variable 

qp1 - qp7 = workplace perception questions, not used in current analysis

Eye-tracking data

aoiname = character variable for descriptive name of each area of interest

criteria_aoi = dummy variable indicating if area of interest is from a criterion

answers_aoi = dummy variable indicating if area of interest is from an answer

degree_aoi = dummy variable indicating if area of interest is from a degree

ttfff_ms = continuous variable indicating the time to first fixations in milliseconds for each area of interest (aoi)

timespent_fms = continuous variable indicating the time in milliseconds spent on each aoi

timespent_f = numeric variable indicating the number of times subjects look at each aoi 

revisit_f_revisits = numeric variable indicating the number of times subjects revisit each aoi

fixations_count = numeric variable indicating the number of times subjects fixate each aoi

first_fixation_duration_ms = continuous variable indicating the length of time in ms of the first time subjects fixate each aoi

average_fixations_duration_ms = continuous variable indicating the avergae length of time in ms of fixations for each subjects

c1 - c16 = dummy variable indicating what of the sixteen criteria the eye-tracking data refers to

d1 - d16 = dummy variable indicating what of the sixteen degrees the eye-tracking data refers to

a1 - a2 = dummy variable indicating what of the two answers the eye-tracking data refers to

aoi_c_d = numeric variable indicating the ordering position of the areas of interest for criteria and degrees together

aoi_c = numeric variable indicating the ordering position of the areas of interest for criteria

aoi_cda = descriptive variable indicating names of areas of interest

---

## [Decision Letter · Decision Letter 2]

16 Nov 2022

Information acquisition and cognitive processes during strategic decision-making: combining a policy-capturing study with eye-tracking data.

PONE-D-22-03769R2

Dear Dr. Pizzo,

We’re pleased to inform you that your manuscript has been judged scientifically suitable for publication and will be formally accepted for publication once it meets all outstanding technical requirements.

Kind regards,

Iván Barreda-Tarrazona, PhD

Academic Editor

PLOS ONE

Additional Editor Comments (optional):

Reviewers' comments:

Reviewer's Responses to Questions

**Comments to the Author**

1. If the authors have adequately addressed your comments raised in a previous round of review and you feel that this manuscript is now acceptable for publication, you may indicate that here to bypass the “Comments to the Author” section, enter your conflict of interest statement in the “Confidential to Editor” section, and submit your "Accept" recommendation.

Reviewer #1: All comments have been addressed

Reviewer #2: All comments have been addressed

2. Is the manuscript technically sound, and do the data support the conclusions?

Reviewer #1: Yes

Reviewer #2: Yes

3. Has the statistical analysis been performed appropriately and rigorously? 

Reviewer #1: Yes

Reviewer #2: Yes

4. Have the authors made all data underlying the findings in their manuscript fully available?

Reviewer #1: Yes

Reviewer #2: Yes

5. Is the manuscript presented in an intelligible fashion and written in standard English?

Reviewer #1: (No Response)

Reviewer #2: Yes

6. Review Comments to the Author

Reviewer #1: (No Response)

Reviewer #2: I had only one comment left in de last round, which has now been addressed. I have no further comments.

7. PLOS authors have the option to publish the peer review history of their article (what does this mean?). If published, this will include your full peer review and any attached files.

Reviewer #1: **Yes: **Joel Huber

Reviewer #2: No

---

## [Editor Report · Acceptance letter]

22 Nov 2022

PONE-D-22-03769R2 

Information acquisition and cognitive processes during strategic decision-making: combining a policy-capturing study with eye-tracking data. 

Dear Dr. Pizzo:

I'm pleased to inform you that your manuscript has been deemed suitable for publication in PLOS ONE. Congratulations! Your manuscript is now with our production department. 

Kind regards, 

on behalf of

Dr. Iván Barreda-Tarrazona 

Academic Editor

PLOS ONE